# Evidence for divergent cortical organisation in Parkinson's disease and Lewy Body Dementia

Angeliki Zarkali [1,2,3] ✉, George Thomas [1], Naomi Hannaway [1], Ivelina Dobreva [1], Melissa Grant Peters [4,5], Martina F. Callaghan [6], Mina Ryten [4,5,7] & Rimona S. Weil [1,2,3]

Dementia is a defining feature of Lewy body disease: its timing and onset distinguish different clinical diagnoses, and its effect on quality of life is profound. However, it remains unclear whether processes leading to cognitive and motor symptoms in Lewy body disease differ. To clarify this, we use in-vivo neuroimaging to assess spatial gradients of inter-regional differences in structural and functional connectivity in 108 people across the Lewy body disease spectrum (46 Parkinson's with normal cognition (PD-NC), 62 Lewy body dementia (LBD)) and 23 controls. We show divergent structural gradient differences with cognitive impairment: PD-NC show increased inter-regional differentiation, whilst LBD show overall gradient distribution similar to controls despite widespread organisational differences at the regional level. We then assess cellular and molecular underpinnings of these organisational changes. We reveal similarities and also important differences in the drivers of cortical organisation between LBD and PD-NC, particularly in layer 4 excitatory neurons.

Lewy body diseases, characterised by the presence of alpha-synuclein aggregates forming Lewy bodies and Lewy neurites, are a group of heterogeneous clinical syndromes including Parkinson's disease (PD), Parkinson's disease dementia (PDD), and Dementia with Lewy bodies (DLB)[1]. Cognitive impairment typically accompanies motor symptoms in Lewy body diseases, and the timing of cognitive impairment defines disease subtypes, with PDD diagnosed when dementia occurs more than a year after motor symptoms and DLB if dementia occurs before[2,3]. PDD and DLB can also be grouped using the term Lewy body dementia (LBD) to refer to both conditions. Dementia, in addition to motor symptoms of PD, is common, with up to 50% of patients with PD developing PDD[4], and males and those with visuospatial impairment are at higher risk[5]. Both PDD and DLB have a devastating impact on

quality of life. Furthering our understanding of how PD with and without dementia differ is essential for disease characterisation and, ultimately, treatment.

Understanding how brain structure is altered in patients with and without dementia is particularly important. Several studies have shown widespread changes in brain structure associated with cognitive decline in Lewy body diseases, including cortical atrophy[6,7] and widespread changes in grey[8–10], and -white matter[11,12] macrostructure. Although these studies provide useful insights, they were unable to account for how different brain regions are interlinked at the cellular and molecular level across large-scale brain networks. This large-scale network architecture is underpinned by fundamental organisational gradients, which represent axes of continuous spatial transitions

[1]Dementia Research Centre, UCL Queen Square Institute of Neurology, University College London, London, UK. [2]National Hospital for Neurology and Neurosurgery, London, UK. [3]Movement Disorders Centre, UCL Queen Square Institute of Neurology, University College London, London, UK. [4]Dementia Research Institute, University of Cambridge, Cambridge, UK. [5]Department of Clinical Neurosciences, University of Cambridge, Cambridge, UK. [6]Department of Imaging Neuroscience, UCL Queen Square Institute of Neurology, University College London, London, UK. [7]Department of Medical Genetics, NIHR Cambridge Biomedical Research Centre, University of Cambridge, Cambridge, UK. ✉e-mail: a.zarkali@ucl.ac.uk

between brain regions[13]. These continuous, gradual changes in macroscale cortical organisation can be captured using eigenvector-based decompositions to cortex-wide similarity measures, or by calculating "cortical organisational gradients". Cortical gradients allow us to link features of brain organisation across different scales and modalities[13,14], and have been shown in health to underlie brain structural[15,16] and functional connectivity[17], accompany brain development[18,19] and evolution[20,21] and also align with the brain's regional cytoarchitecture and gene expression[16,22]. Cortical gradients capture continuous cortical hierarchical changes[13,23] and are thus more sensitive to global changes occurring in the presence of disease. Figure 1 illustrates how cortical gradients reflect inter-regional changes across different dimensions and how this cortical organisation can change in disease.

Differences in regional cortical organisation gradients are seen in psychiatric and neurological disease, with functional alterations in depression[24] and schizophrenia[25,26] and structural alterations in epilepsy[27]. These axes of organisation may also be relevant to cognition, with structural gradient expansion, or increased spread between extremes of the cortical gradients (as in Fig. 1C), reflecting higher inter-regional differentiation between unimodal and more transmodal regions in adolescence, supporting executive function[28]. In LBD, where widespread, rather than focal alterations are commonly seen[6–12], assessing whether cortical gradients differ across the spectrum of the disease may be particularly relevant. Gradients also offer a framework to explore underlying cellular and molecular correlates of such organisational changes[13,23].

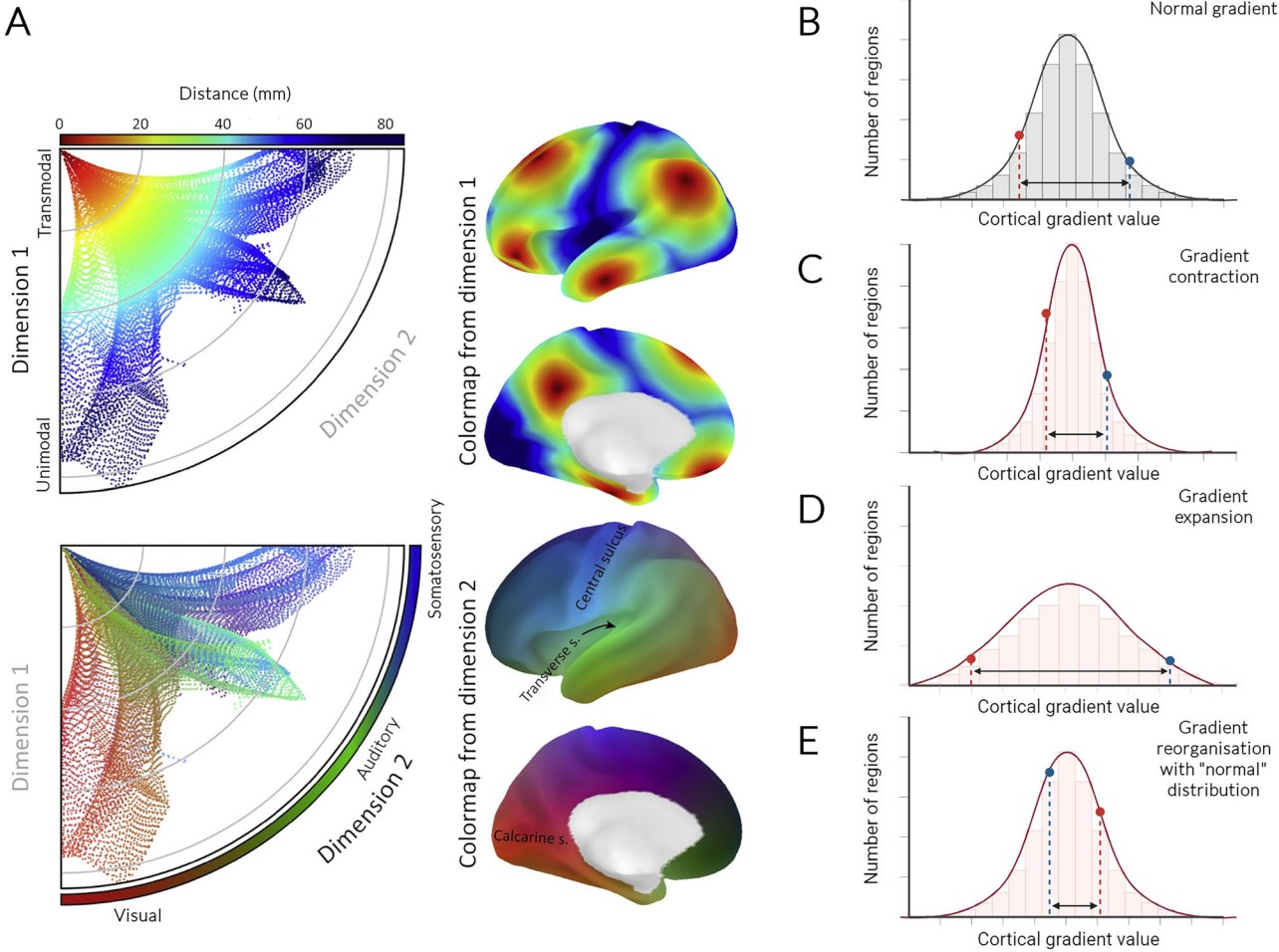

**Fig. 1 | Conceptualisation of cortical gradients and potential alterations in the presence of disease. A** Cortical gradients reflect continuous changes on different dimensions across the brain. Adapted from Huntenburg et al.[14] under a Creative Commons CC-BY license with no changes made: https://creativecommons.org/licenses/by/4.0/. Top: Data points are coloured along the first dimension, which is given as the distance between unimodal (sensorimotor) and transmodal regions (higher on the cortical hierarchy) along the cortical surface. When displayed on the cortical surface, it aligns with a unimodal-to-transmodal cortical gradient. For any two regions, the difference in their gradient value represents how different these regions are in this dimension. Bottom: Data points are coloured according to the second dimension, which differentiates in this case between different sensory modalities. The position of each brain region along this dimension reflects its relative distance across three morphological landmarks (calcarine sulcus, transverse sulcus, central sulcus). **B**–**E** Conceptualisation of different possible changes in gradient distribution in the presence of disease. The distribution of gradient values of a hypothetical structural connectivity gradient in healthy humans is shown in (**B**). The distance between two brain regions (red and blue dots) reflects how different these brain regions are in their structural connectivity patterns along that cortical gradient dimension. In the presence of neurodegeneration, changes in structural connectivity may lead to alterations in gradient values between regions in different directions. *Gradient contraction* (**C**) reflects a narrower distribution with more regions having gradient values along the mean. In this case, the difference between brain regions (the line between the red and blue dots) is reduced. In contrast, widening of the overall gradient distribution, or *Gradient expansion* (**D**), with widening of extremes of gradient values, reflects increased differences between brain regions (red and blue dots) in terms of their structural connectivity profiles. Finally, whilst the overall distribution may remain similar to controls, there may be *Gradient reorganisation* (**E**) in the presence of disease. In this case, brain regions change in their gradient value, but the overall distribution remains the same. Created in BioRender. Zarkali (2025) https://BioRender.com/gqydsn.

We previously showed that differences in structure-function coupling in patients with PD at higher risk of dementia follow known cortical organisational gradients[29]. However, no imaging studies have yet examined whether cortical reorganisation accompanies cognitive decline in Lewy body diseases, nor the spatial pattern of these changes. Recent evidence from post-mortem gene expression suggests that cortical organisation is disrupted in both PD and LBD, but in divergent ways[30]. Single-cell RNA sequencing from two cortical regions with different alpha-synuclein burden found increased intra-individual differences in PD but attenuation of regional identity in PDD, along with distinct cellular and molecular underpinnings of this reorganisation[30]. It remains unknown whether such differences extend across the whole brain, and how they differ between PD with intact cognition (PD-NC) and LBD has yet to be established.

Our goal was to answer this question in vivo, assessing how large-scale differences in cortical structural and functional organisation differ between patients with LBD, PD-NC and unaffected controls (HC) (Fig. 2) and explore associations with disease severity. Cortical gradients provide an ideal framework to assess this as they offer a measure of inter-regional differentiation sensitive to global organisational changes[13,23]. We hypothesised that PD-NC would show increased spread between extremes of the gradient distribution (or overall gradient expansion, Fig. 1D) reflecting more different connectivity profiles between regions[27,28], or increased inter-regional differentiation; whilst LBD would show similar overall distribution compared to controls but changes at regional and local levels (similar to Fig. 1E). Additionally, we hypothesised that there would be differences in the underlying mechanisms of these changes between PD-NC and LBD patients.

To do this, we first compared both the overall distribution and vertex-wise differences in in vivo structural and functional gradients, derived from diffusion-weighted and resting-state functional MRI (rsfMRI), between LBD, PD-NC and HC. We found widening of structural gradients, indicating increased inter-regional differentiation in PD-NC but apparent normalisation of inter-regional differentiation in LBD compared to controls. However, there were widespread differences in regional gradient rankings in LBD for the primary structural gradient (SC-G1), suggesting significant local and regional reorganisation despite normalisation of the overall distribution. We provide further evidence for these findings using region-of-interest (ROI) analysis and 7T quantitative MRI (qMRI) in a separate cohort. Specifically, we assessed whether regions at the extremes of SC-G1 rankings differed more or less in their connectivity profiles between LBD, PD-NC and controls. We confirmed that inter-regional differences in myelin-sensitive qMRI significantly differed between groups. Then, we linked these structural changes to disease severity by correlating the overall difference from control SC-G1 ratings with cognitive and motor severity measures. Structural organisation was behaviourally relevant in LBD and specific to cognitive severity. Finally, we examined whether disease-related gradient alterations corresponded to normative topographical variations in excitatory and inhibitory neurons, cortical cytoarchitecture and global gene expression. We reveal similarities but also important differences in the drivers of changes in cortical organisation between LBD and PD-NC, particularly in excitatory neurons.

## Results

The overall study methodology is seen in Fig. 2. 62 LBD patients (including DLB, PDD and PD with mild cognitive impairment (PD-MCI)), 46 PD patients with stable cognition (PD-NC) and 23 healthy age-matched controls (HC) were included. Demographics and clinical assessments are seen in Table 1. Full cognitive assessments and characteristics of different LBD subgroups are seen in Supplementary Tables 1 and 2.

### Structural connectivity gradients are distinct in LBD and PD-NC
First, we derived cortex-wide structural and functional connectivity gradients for each participant. To assess how inter-regional gradients

differ in LBD and PD-NC, we then estimated eigenvectors describing spatial gradients of systematic cortical variation in brain structural and functional connectivity, also known as cortical gradients. We compared structural and functional gradients between controls, PD-NC and LBD. First, we assessed the overall gradient distribution for the primary structural connectivity gradient (SC-G1) and the second structural connectivity gradient (SC-G2). We found that PD-NC patients showed expansion (scores shifting away from the midpoint or increased spread between extremes of the distribution) of structural connectivity gradients compared to controls and LBD (SC-G1: Kruskal–Wallis $H = 7.87$, $p = 0.019$; SC-G2: $H = 17.0$, $p < 0.001$). This suggests heightened dissimilarity in structural connectivity profiles between regions in PD-NC patients, or increased inter-regional differentiation between different brain regions. In contrast, LBD patients showed constriction of the overall gradient distribution compared to PD-NC, with a slight qualitative reduction of regions in the extreme of gradients compared to controls (SC-G1: Fig. 3A; *SC-G2:* Supplementary Fig. 1). This suggests that increased inter-regional differentiation is a feature of PD-NC but not LBD.

Next, to better understand regional contribution to global cortical organisation and how a given region's position within the cortical hierarchy may differ in Lewy body diseases, we used surface-based mixed linear models (age and sex as covariates, FWE-correction for multiple comparisons, cluster threshold 0.01), with each vertex allocated to its corresponding structural gradient value. Differences in gradient values in PD-NC compared to controls were concentrated in the extremes of the gradient distribution for both gradients: for SC-G1, changes primarily involved unimodal regions such as the sensorimotor and left visual cortex (Fig. 3B). For SC-G2 there were reductions in gradient values of left prefrontal regions (Supplementary Fig. 1). In contrast, and despite a similarity of overall gradient distribution, there were bidirectional, bilateral and widespread differences in their spatial organisation in LBD compared to controls for both structural connectivity gradients, SC-G1 (Fig. 2C) and SC-G2 (Supplementary Fig. 1). SC-G1 differences in LBD compared to controls were similar when assessing different LBD subgroups separately (Supplementary Fig. 2).

When directly comparing PD-NC and LBD at the vertex rather than region-of-interest (ROI) level, we found no statistically significant differences (uncorrected results in Supplementary Fig. 3 and differences between LBD subgroups and PD-NC in Supplementary Fig. 4). In contrast to the pronounced differences in structural connectivity gradients shown in our cohort, functional connectivity gradients did not differ between any groups (Supplementary Fig. 5).

Several replication analyses (with structural connectivity gradients derived using different sparsity thresholds) demonstrated the robustness of structural gradient differences in LBD and PD-NC compared to controls (Supplementary Fig. 6).

### Differences in inter-regional differentiation between LBD and PD-NC are also seen using ultra-high field quantitative MRI in a different cohort of LBD, PD-NC and controls
To ensure the robustness of our results, we provide further evidence in support of our findings in a separate cohort of LBD, PD-NC and controls using different MR acquisitions and analyses. We used 7 Tesla quantitative MRI (7 T qMRI) to test qMRI values from regions at the extremes of SG-G1 distribution (that differed between groups in our main analysis) against regions from the middle of the distribution (which did not differ between groups).

We included 13 LBD, 13 PD-NC and 23 age-matched controls in our 7 T cohort. Demographics and results of clinical assessments in the 7 T cohort are presented in Supplementary Table 3. We computed quantitative multiparametric maps (MPM) including proton density, longitudinal relaxation rate (R1), effective transverse relaxation rate (R2*), and magnetisation transfer saturation (MTsat). We extracted mean

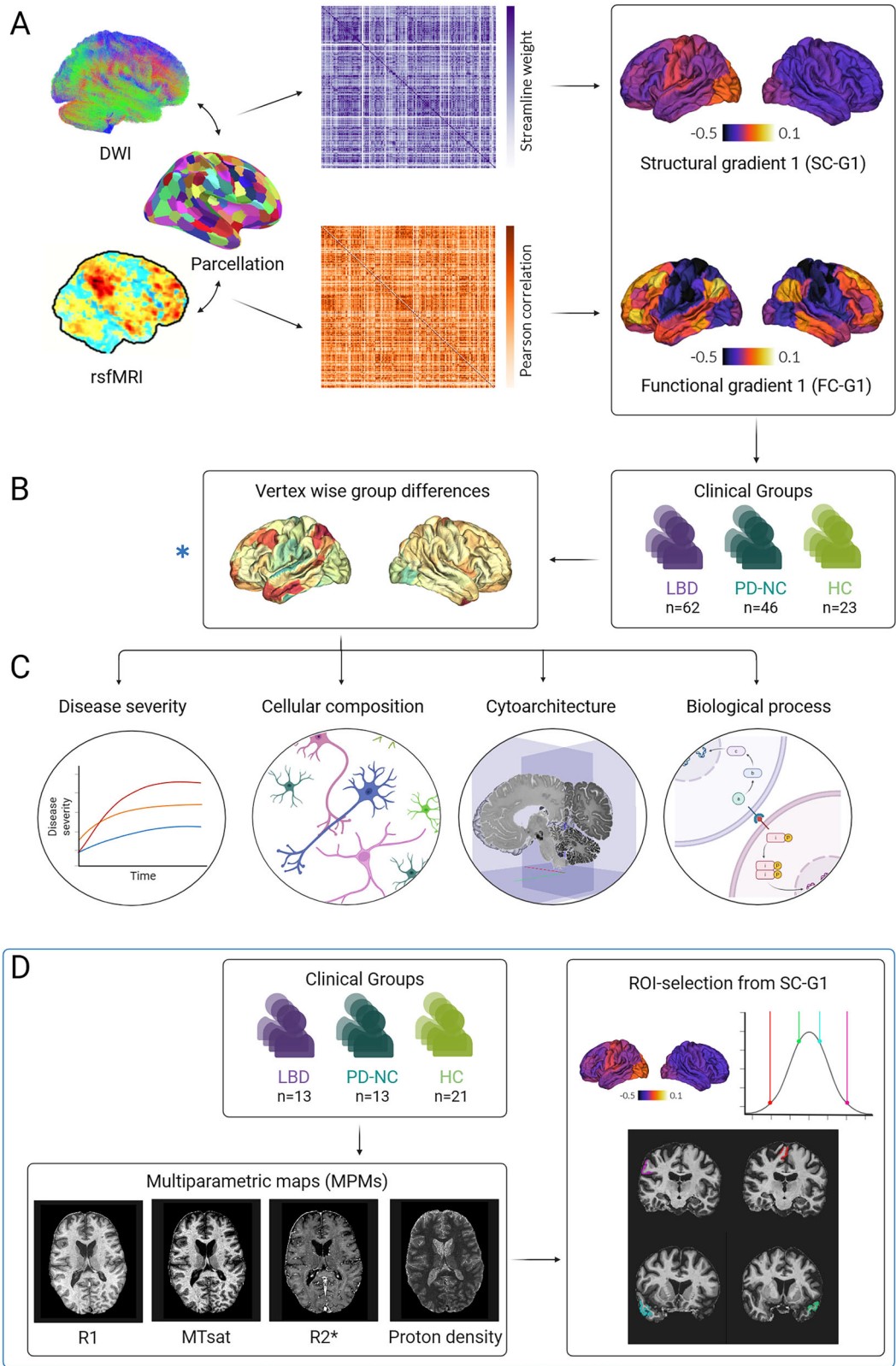

MPM values for four ROIs of the Schaeffer parcellation based on their SC-G1 ratings from the main 3 T cohort and whether they differed in LBD compared to controls. Thus, two regions were selected from the extremes of the gradient distribution, which differed between LBD and controls ("RH_SalVentAttn_TempOccPar_3", and "RH_SomMot_18") and two were selected from the middle of the gradient distribution and which did not show differences between LBD vs controls ("RH_Default_Temp_1", "LH_Default_Temp_1"). We then used mixed linear models accounting for age and sex to study the ROI*Group (LBD, PD-NC, HC) interaction and assess whether inter-regional differences in MPM values differed significantly between groups. We were particularly interested in myelin-sensitive markers (MTsat and R1), given our

**Fig. 2 | Assessing cortical organisation across the spectrum of Lewy body disease. A** Individualised structural and functional gradient construction. Diffusion-weighted (DWI) and resting-state functional MRI (rsfMRI) images were obtained from each participant. Structural T1-weighted imaging was used to parcellate the brain into 200 cortical regions of interest using the Schaeffer parcellation[87]. The same parcellation was used to derive structural connectivity matrices weighted by streamlines between any two regions (purple) and functional connectivity matrices, weighted by Pearson correlation between any two regions (orange), for each participant. We then applied diffusion map embedding, a non-linear dimensionality reduction technique[17,88] to identify gradients (eigenvectors) of the main spatial axes in inter-regional similarity of structural and functional connectivity. The average principal gradient for structural (SC-G1) and functional connectivity (FC-G1) in healthy controls is shown. **B** Surface-based linear models revealed significant differences in structural gradient scores (SC-G1) between groups. Gradients were compared between 3 groups: 62 patients with Lewy body dementia (LBD), 46 patients with Parkinson's disease with intact cognition (PD-NC) and 23 unaffected controls (HC). We used surface-based linear models controlling for age and sex to compare gradient scores between LBD vs HC and PD-NC vs HC. We validated changes in structural organisation using a region-of-interest analysis (in regions across the SC-G1 spectrum) and quantitative ultra-high field 7 Tesla MRI in a separate cohort of patients. **C** Neural contextualisation of cortical structural organisation alterations. We assessed whether the (1) the structural SC-G1 gradient alterations seen in Lewy body disease were related to cognitive and motor disease severity and whether (2) the spatial distribution of vertex-wise SC-G1 alterations seen in Lewy body disease (LBD vs HC and PD vs HC) was correlated with the spatial distribution of specific neuronal and glial cell types, cytoarchitectural axes of organisation, and expression of genes linked to specific biological processes and pathways. Created in BioRender. Zarkali (2025) https://BioRender.com/ylylz18. **D** Replication analysis in a separate cohort using 7 Tesla quantitative MRI. Multi-parametric maps (MPMs) were acquired at 7 Tesla for each participant, including maps sensitive to myeline such as longitudinal relaxation rate (R1) and magnetisation transfer saturation (MTsat), iron-sensitive map effective transverse relaxation rate (R2*), and a map sensitive to overall water content, proton density. We then extracted the mean MPM signal for four regions of interest (ROIs) of the same Schaeffer parcellation used in our main analysis. ROIs were selected based on their SC-G1 ratings from the main 3 T cohort and whether they differed in LBD compared to controls. Thus, two regions were selected from the extremes of the gradient distribution, which differed between LBD and controls ("RH_SalVentAttn_TempOccPar_3", and "RH_SomMot_18") and two were selected from the middle of the gradient distribution and which did not show differences between LBD vs controls ("RH_Default_Temp_1", "LH_Default_Temp_1"). We assessed for ROI*Group interaction using mixed linear models with age and sex as covariates to assess whether inter-regional differences in MPM signal differed significantly between the three clinical groups.

## Table 1 | Demographics and clinical characteristics

| Characteristic | Controls (HC) n = 23 | Parkinson's with normal cognition (PD-NC) n = 46 | Lewy body dementias (LBD) n = 62 | p-value |
|---|---|---|---|---|
| Age | 66.8 (8.6) | 62.3 (7.2) | 70.1 (6.7) | **p = 0.030[b]** |
| Male (%) | 11 (47.8) | 22 (47.8) | 50 (80.6) | **p < 0.001[a,b]** |
| Right-handed (%) | 21 (91.3) | 45 (97.8) | 53 (85.5) | p = 0.190 |
| Years of education | 17.6 (2.4) | 17.0 (2.6) | 16.3 (3.7) | p = 0.163 |
| MOCA | 28.6 (2.6) | 28.9 (1.2) | 23.4 (5.4) | **p < 0.001[a,b]** |
| MMSE | 28.9 (1.9) | 29.2 (1.2) | 26.6 (3.5) | **p < 0.001[a,b]** |
| HADS anxiety | 3.8 (3.7) | 4.6 (2.7) | 6.6 (4.7) | p = 0.012[a] |
| HADS depression | 2.6 (2.9) | 3.5 (2.4) | 6.2 (3.1) | **p < 0.001[b]** |
| Composite cognitive score | - | 0.1 (0.5) | −1.8 (1.7) | **p < 0.001[a,b]** |
| Years of diagnosis | - | 4.7 (2.9) | 4.0 (4.0) | p = 0.090 |
| UPDRS total | - | 49.9 (15.5) | 66.3 (28.5) | **p = 0.008** |
| UPDRS part 3 | - | 28.3 (8.3) | 32.8 (15.6) | p = 0.307 |
| Timed Up and GO | - | 8.9 (1.9) | 10.3 (4.5) | p = 1.000 |
| Functional Assessments Questionnaire | - | 1.7 (1.9) | 8.3 (6.4) | **p < 0.001** |
| CAF | - | 2.4 (3.1) | 4.7 (3.7) | p = 0.171 |
| One-Day Fluctuations Questionnaire | - | 1.4 (2.0) | 6.4 (6.9) | **p = 0.020** |
| LEDD | - | 500.0 (389.7) | 515.9 (253.8) | p = 3.719 |
| UMPDHQ | - | 0.8 (1.9) | 2.9 (3.3) | **p < 0.001** |
| RBDSQ | - | 4.5 (2.8) | 7.0 (4.0) | **p = 0.012** |

Characteristics were compared using ANOVA (post-hoc Tukey) for continuous normally distributed variables and Kruskal–Wallis (post-hoc Dunn) for non-normally distributed variables.
In bold, significant differences between LBD and PD. All results shown as mean (standard deviation) except otherwise indicated.
*CAF* Clinician Assessment of Fluctuations, *HADS* Hospital Anxiety and Depression Scale, *LEDD* Levodopa Equivalent Daily Dose, *MMSE* Mini-Mental State Examination, *MOCA* Montreal Cognitive Assessment, *RBDSQ* REM Sleep Behaviour Disorder Sleep Questionnaire, *UMPDHQ* University of Miami PD Hallucinations Questionnaire, *UPDRS* Movement Disorders Society Unified Parkinson's Disease Rating Scale.
[a]Difference between LBD-HC.
[b]Difference between LBD-PD.

gradient changes were seen in structural connectivity gradients. We found a significant ROI*Group interaction for myelin-sensitive maps, both R1 ($p = 0.045$) and MTsat ($p = 0.022$), but not for proton density or R2*. For MTsat, the group difference in inter-regional MTsat values was driven by the LBD group in the "RH_SomMot_18" region ($\beta = -0.131$, $p = 0.003$). For R1, there were no significant regions or groups driving the overall ROI*Group interaction in pairwise comparisons, suggesting the overall difference was not driven by a single region. Full results of the 7 T analysis are presented in Supplementary Fig. 7.

## Structural connectivity organisation changes are behaviourally relevant and specific to cognitive performance and Lewy body pathology

Next, we examined whether the changes in structural cortical organisation in Lewy body disease were related to disease severity. To do this, we calculated for each participant a composite gradient difference score, derived from the absolute Z-scored SC-G1 gradient value for each region compared to the control gradient values for that region, then summed across the 200 regions; this way, higher scores for a participant reflect more differences in overall cortical organisation

## Structural connectivity gradient SC-G1 organisational differences

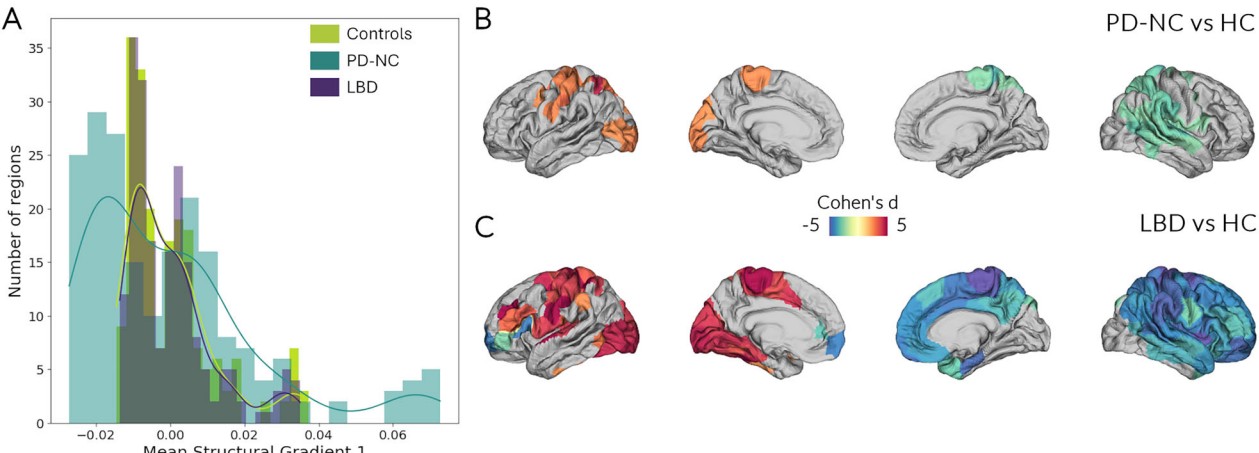

## Gradient organisational differences relate to cognitive but not motor severity in LBD

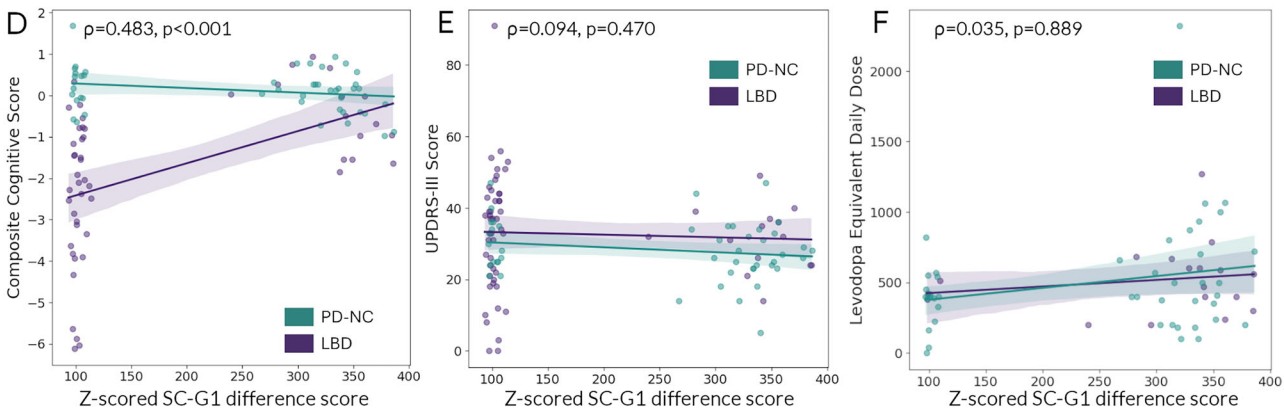

**Fig. 3 | Structural connectivity gradient alterations are seen in Lewy body disease and are specific to cognitive severity.** Overall gradient distribution in the principal structural gradient (SC-G1) differed between controls (HC), Parkinson's patients with normal cognition (PD-NC) and patients with Lewy Body Dementias (LBD) (**A**), with expansion of gradient scores in PD-NC reflecting increased inter-regional differentiation in structural connectivity (Kruskal–Wallis $H = 7.87$, $p = 0.019$). Regional differences in structural gradient scores were seen in both PD-NC (**B**) and LBD patients compared to HC (**C**). Surface-based linear models controlling for age and sex (family-wise error correction for multiple comparisons, cluster threshold 0.01) revealed significant differences in SC-G1 gradient scores between groups. Only statistically significant clusters after multiple comparisons correction ($p_{FWE} < 0.05$) are shown. Colour scale depicts effect size per vertex (blue colours decrease in gradient scores; red colours increase in gradient scores). Gradient reorganisation is specific to cognitive severity in LBD. For each participant, a composite gradient change score was calculated from the Z-scored gradient value for each region compared to HC gradient values for that region, summed across the 200 regions. Composite gradient scores were correlated with composite cognitive scores ($p < 0.001$) (**D**) but not motor scores ($p = 0.470$) (UPDRS-III: Unified Parkinson's Disease Rating Scale Part 3) (**E**) or levodopa daily equivalent doses ($p = 0.889$) (**F**). Spearman correlation was used; error bars represent 95% confidence intervals. Source data are provided as a Source Data file.

than controls. We correlated composite gradient difference scores with measures of disease severity (cognitive severity: Montreal Cognitive Assessment (MOCA) and mini-mental state assessment (MMSE), and a composite cognitive score combining detailed cognitive assessments across 5 cognitive domains); and motor severity: Movement Disorders Society Unified Parkinson's Disease Scale part III (MDS-UPDRS-III) and "timed up and go" (TUG) score within each disease group (LBD and PD-NC). We found that having gradient scores closer to controls (lower composite gradient difference scores) was related to worse cognitive performance within LBD patients (SC-G1: MOCA: Spearman $\rho = 0.234$, $p = 0.062$; MMSE: $\rho = 0.309$, p = 0.015; composite cognitive score: $\rho = 0.483$, $p < 0.001$) but not within PD-NC (all non-significant; Fig. 3D). This further supports the idea that apparent similarity in inter-regional differentiation between control and LBD participants is in fact associated with behaviourally relevant changes in organisation at the regional level. Indeed, participants with more "apparently normal" gradients are those with worse cognition. Correlation with gradient scores remained significant when correcting for

age, sex and p-tau-217 values for combined cognitive scores ($p = 0.001$), MOCA ($p = 0.050$), but not MMSE ($p = 0.272$).

In contrast there was no correlation between composite gradient difference scores and motor measures (MDS-UPDRS-III or TUG scores, Fig. 3E) or levodopa equivalent daily dose (LEDD) (Fig. 3F). These findings suggest that the differences in structural cortical organisation seen in LBD and PD-NC patients are specific to cognitive decline in Lewy body diseases, rather than other clinical features such as motor severity.

### Neuronal underpinnings of structural gradient differences show divergent contributions in PD-NC and LBD, with excitatory neurons affected only in LBD

We then aimed to evaluate whether structural gradient differences are associated with regional variations in specific neuronal and glial cell populations. To do this, we correlated the unthresholded t-map of gradient differences in LBD vs controls and PD-NC vs controls with regional gene expression of cell-specific gene markers (against 1000 spatially correlated spin permutations). We then corrected for

multiple comparisons across all comparisons for each t-map ($q_{spin}$: FDR-corrected $p_{spin}$). We found that SC-G1 differences in LBD compared to controls were associated with the regional distribution of inhibitory neuronal cells ($\rho = -0.417$, $p_{spin} < 0.001$, $q_{spin} = 0.004$), and oligodendrocytes ($\rho = -0.193$, $p_{spin} = 0.021$, $q_{spin} = 0.042$). We further examined whether specific cell types were driving the correlation with excitatory or inhibitory neuronal cell populations. SC-G1 differences in LBD were correlated with RORB-expressing layer 4 neurons ($\rho = 0.257$, $p_{spin} = 0.022$, $q_{spin} = 0.042$), with no other excitatory neuronal marker showing significant correlation. The relationship with inhibitory cell distribution was driven by PAX6 and SST inhibitory neurons. These findings suggest that regions differentially expressing inhibitory and excitatory cell populations are more likely to show changes in cortical hierarchical organisation in LBD; with regions rich in excitatory and poor in inhibitory cells (higher E-I ratio, particularly in RORB layer 4 neurons) showing increased SC-G1 values and regions with lower E-I ratio and poorer in oligodendrocytes showing contraction in SC-G1 values (Fig. 4A).

In contrast, regional changes in gradients in PD-NC compared with controls were only correlated with inhibitory (total, VIP and SST) but not excitatory neuronal distribution, and with oligodendrocytes ($\rho = 0.199$, $p_{spin} = 0.018$, $q_{spin} = 0.049$); regions richer in excitatory neurons and poorer in oligodendrocytes were more likely to show contraction in SC-G1 values in PD-NC. Full results of neuronal underpinnings of structural gradient alterations in LBD and PD-NC are shown in Table 2.

### Microarchitectural underpinnings of structural gradient alterations

Next, we examined whether disease-related differences in the principal structural connectivity gradient reflect large-scale differences in cortical microstructure. We correlated changes in SC-G1 values (t-maps) in LBD vs controls and PD-NC vs controls to atlases of normative cytoarchitectural organisation and laminar thickness.

We found that primary gradient (SC-G1) alterations were spatially correlated with sensory-fugal cytoarchitectural differentiation derived from normative histology data, or a gradient of differentiation from more unimodal sensory areas to more transmodal cortical areas (LBD vs HC gradient changes: $\rho = -0.369$, $q_{spin} = 0.009$; PD-NC vs HC: $\rho = 0.295$, $q_{spin} = 0.004$). This was replicated by using a unimodal-to-transmodal ranking based on the functional network allocation for each region (LBD: $\rho = -0.352$, $p_{spin} = 0.001$; PD-NC: $\rho = -0.352$, $p_{spin} = 0.001$), with more transmodal regions more likely to have increased gradient scores and more unimodal regions likely to have contracted gradient scores. In contrast, there was no correlation with an anterior to posterior axis of cytoarchitectural differentiation (Fig. 4B).

Differences in structural connectivity organisation in LBD and PD-NC also showed specific laminar contributions, with both SC-G1 gradient change t-maps showing negative correlations with the regional distribution of layer 1 and positive correlations with layer 4 thickness (Table 2). This suggests that regions with more reduced SC-G1 values in those groups are more likely to have a thicker layer 1 and a thinner layer 4. SC-G1 gradient changes in LBD but not in PD-NC were also correlated with layer 6 thickness (Table 2), suggesting regions with more reduced SC-G1 values in LBD are more likely to have thicker layer 6.

### Structural gradient alterations are disease-specific and not driven by Alzheimer's pathology

To assess whether the differences in structural cortical organisation we saw in LBD participants were driven by Alzheimer's co-pathology or were disease-specific, we performed two analyses. First, we assessed whether composite gradient difference scores were correlated to plasma p-tau217 levels (a plasma biomarker that correlates to brain levels of beta-amyloid and tau on PET[31]). We found no correlation for SC-G1 ($\rho = 0.132$, $p = 0.346$) or SC-G2 ($\rho = 0.172$, $p = 0.217$).

Secondly, we examined whether LBD-related differences in SC-G1 (unthresholded t-map of SC-G1 changes between LBD vs HC) were correlated with regional expression of mendelian risk genes for PD and AD, using spatially correlated spin permutations. We found that LBD gradient changes were correlated with expression of PD-specific genes ($\rho = -0.256$, $p_{spin} = 0.039$) but not AD genes ($\rho = -0.046$, $p_{spin} = 0.549$). Together, these findings suggest that Alzheimer's co-pathology does not drive the gradient alterations seen in LBD participants.

### Structural gradient differences are associated with variation in regional gene expression patterns: the same regions drive this in LBD and PD-NC, but there are differences in biological processes and pathways

Finally, we wanted to assess whether principal structural gradient alterations in LBD and PD-NC are underpinned by normative differences in genes related to specific biological processes and pathways. To do this, we performed partial least squares regression (PLS) with dependent variable Y, the t-map of SC-G1 gradient alterations (1*200 regions, LBD vs HC and PD-NC vs HC) and predictor matrix X regional gene expression (17545*200 regions from the Allen Human Brain Atlas[32]); we tested our results against 10000 permutations of spatially correlated spin permutations of the gradient t-map. The first principal component, PLS1, explained the most variance in both gene expression and gradient alteration variability (Supplementary Table 4) and therefore, PLS1 gene weightings were used for further gene ontology and enrichment analyses (only for genes significantly differentially weighted $p_{spin} < 0.05$).

For both LBD and PD-NC compared to controls, we observed that shifts in cortical gradients were associated with down-weighting, or reduced expression of specific genes: 2782 genes in LBD and 2296 in PD-NC. PLS1 region weights were positively correlated with gradient alterations seen in LBD ($\rho = 0.389$, $p_{spin} < 0.001$). In other words, areas where gradient scores differed most in LBD had lower expression of genes with the most negative PLS1 weights. The regional profile of PLS1 weighting for both LBD and PD-NC was similar, with differences in regional expression of the same left parietal and frontal regions in both patient cohorts (Supplementary Fig. 9A).

Downweighted genes, which were less expressed in regions showing differences in LBD vs controls, were enriched for GO terms relating to cellular response to stimulus, positive regulation of biological and cellular processes, and developmental processes (Supplementary Fig. 9). They were also enriched for REACTOME pathways relating to lipid-associated metabolic pathways (Tables 3 and 4).

We found a similar pattern for Gene Ontology (GO) terms and REACTOME pathways linked to downweighted genes in PD-NC compared to controls (Supplementary Fig. 9). The ranking of GO terms for LBD vs HC and PD-NC vs HC was highly inter-correlated ($\rho = 0.807$, $p < 0.001$) with greater overlap than expected by chance (hypergeometric test $p < 0.001$). The same was true for REACTOME pathways ($\rho = 0.733$, $p = 0.025$, hypergeometric test $p < 0.001$). However, there were several terms exclusively enriched in one disease group (Supplementary Fig. 9). 33 terms were only enriched in the PD-NC downweighted gene set and primarily involved signal transduction and response to stimulus. Whilst 62 terms were uniquely enriched in the LBD gene set and related to positive regulation of cellular processes, developmental processes and catabolic processes. Full tables of GO terms are seen in Supplementary Table 5.

## Discussion

Here, we comprehensively assessed cortical organisation differences in Lewy Body Diseases and how these changes relate to underlying cellular and molecular processes. We showed that differences in PD-NC and LBD diverged. In PD-NC, we found expansion of cortical gradients,

## Gradient differences in LBD but not PD-NC relate to RORB excitatory neurons

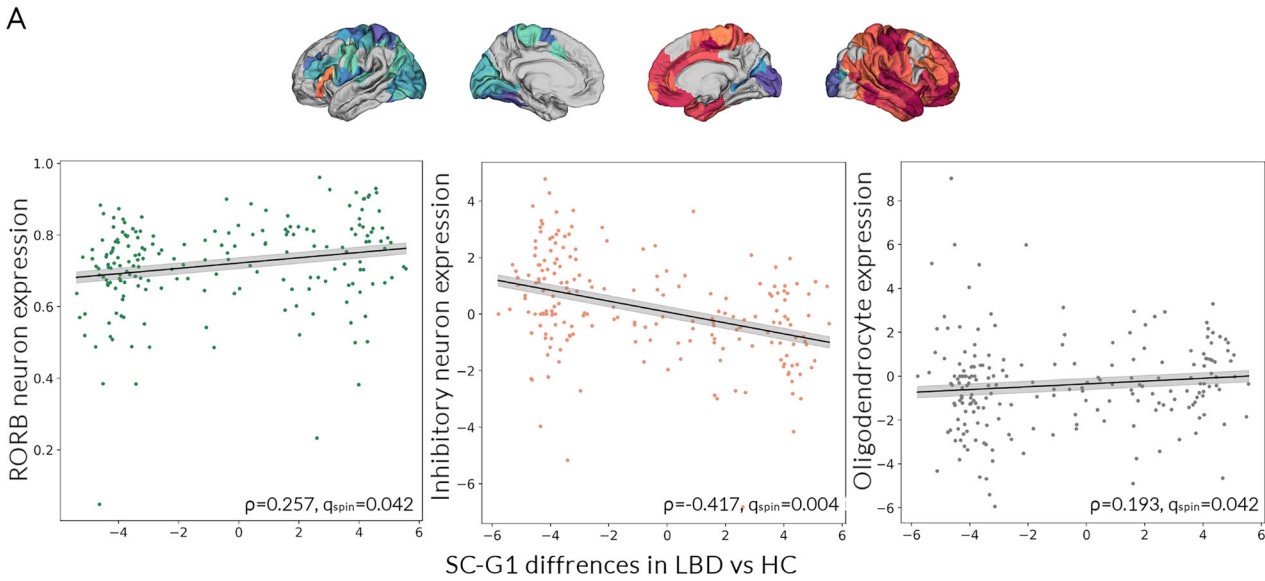

## Gradient differences relate to cytoarchitecture and laminar thickness

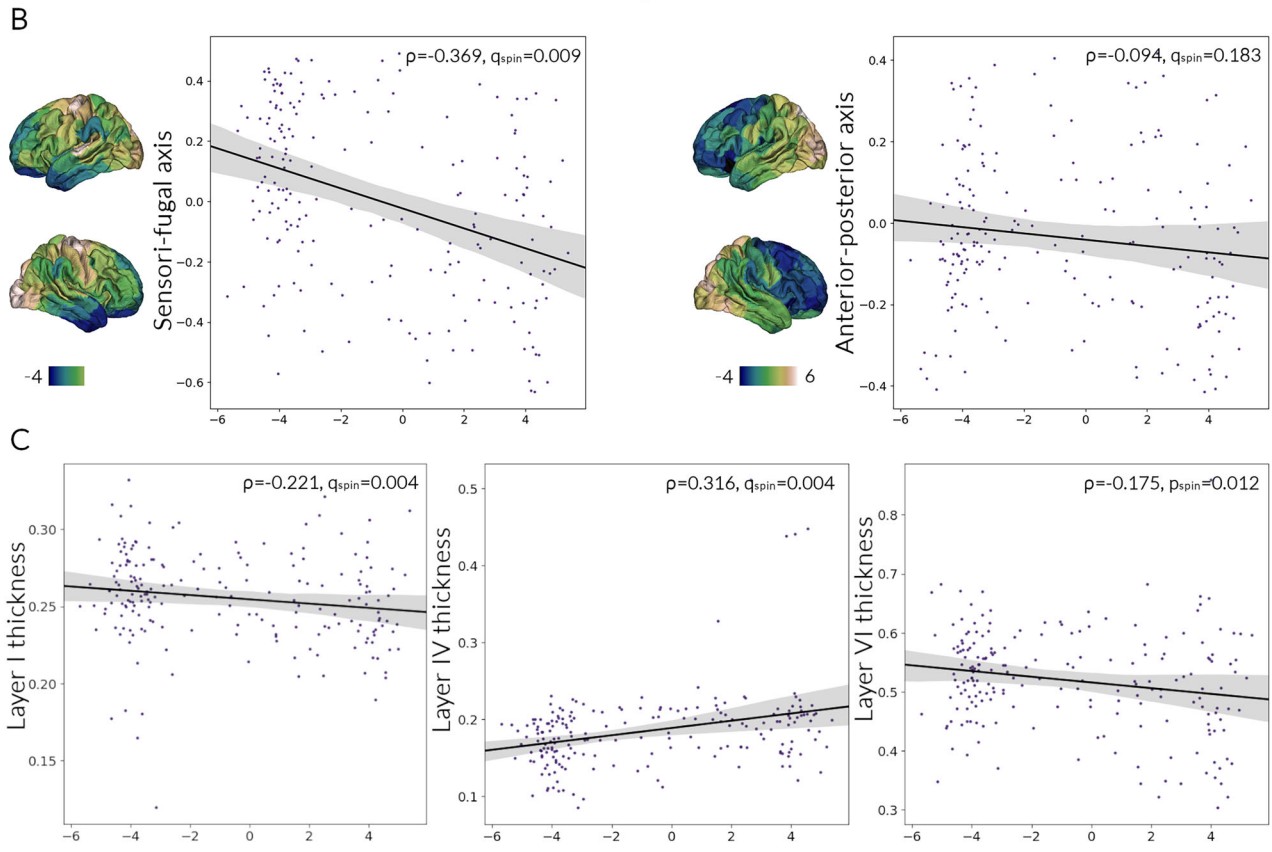

suggesting an exacerbation of inter-regional differences. In contrast, in LBD, we found attenuation of inter-regional differentiation with overall gradient distribution consistent with controls; however, this was accompanied by widespread differences in regional cortical gradient values in LBD patients. We show that these changes are relevant specifically to cognitive but not motor severity. Importantly, we relate these differences in cortical hierarchies to normative cell type composition, cortical cytoarchitecture and gene expression. We found that cortical reorganisation in both PD-NC and LBD patients is more pronounced in regions normally richer in inhibitory neurons and poorer in oligodendrocytes. However, only in LBD patients were differences correlated with excitatory neurons, specifically RORB-expressing layer 4 neurons. Although PD-NC and LBD showed similar regional patterns and biological pathways driving reorganisation, key

**Fig. 4 | Neuronal underpinnings of the structural gradient alterations in LBD patients. A** The spatial distribution of structural gradient alterations in Lewy body dementia participants (LBD) is significantly correlated with the spatial distribution of normative gene expression markers of RORB excitatory neurons, inhibitory neurons and oligodendrocytes. The unthresholded t-map of primary structural gradient (SC-G1) changes in LBD vs healthy controls (HC) was correlated with normative regional gene expression of cell-specific gene markers (against 1000 spatially correlated spin permutations) from the Allen Human Brain Atlas[32]. *P*-values presented are corrected for multiple comparisons using false discovery rate across all comparisons per group (q$_{spin}$). SC-G1 changes in LBD compared to HC were positively associated with the regional distribution of RORB-expressing layer 4 neurons (left) and overall inhibitory neuronal cells (middle) as well as

oligodendrocytes (right). Error bars represent 95% confidence intervals.
**B** Structural gradient alterations in LBD follow a sensori-fugal axis of cytoarchitecture. The SC-G1 t-map of changes in LBD vs HC was correlated to atlases of normative cytoarchitectural organisation from BigBrain[99] (against 1000 spatially correlated spin permutations). SC-G1 alterations were spatially correlated with the sensory-fugal but not the anterior-posterior axis of cytoarchitectural differentiation. Error bars represent 95% confidence intervals. **C** Structural gradient alterations in LBD show specific laminar contributions SC-G1 gradient change t-maps (LBD vs HC) were negatively correlated with the regional variation of layer 1 (left) and layer 6 (right) cortical thickness and positively correlated with layer 4 cortical thickness (middle). Error bars represent 95% confidence intervals. Source data are provided as a Source Data file.

## Table 2 | Neuronal and microstructural underpinnings of structural gradient 1 (SC-G1) alterations in Lewy Body Diseases

| Characteristic | Parkinson's with normal cognition (PD-NC) | | Lewy body dementias (LBD) | |
|---|---|---|---|---|
| Cell-specific gene markers | ρ | q$_{spin}$ | ρ | q$_{spin}$ |
| **Excitatory neurons*** | 0.147 | 0.111 | 0.192 | 0.060 |
| CUX2 | −0.069 | 0.773 | −0.013 | 0.699 |
| THEMIS | 0.059 | 0.509 | 0.069 | 0.504 |
| RORB | 0.179 | 0.098 | **0.257** | **0.042** |
| FEZF2 | 0.003 | 0.773 | −0.059 | 0.504 |
| **Inhibitory neurons*** | **−0.376** | **0.019** | **−0.417** | **0.004** |
| PAX6 | **−0.165** | 0.085 | **−0.226** | **0.034** |
| RELN | **−0.244** | 0.063 | −0.249 | 0.060 |
| VIP | **−0.355** | **0.013** | −0.308 | 0.062 |
| SV2C | 0.138 | 0.169 | 0.141 | 0.348 |
| TLE4 | −0.022 | 0.698 | −0.031 | 0.592 |
| PVALB | 0.167 | 0.098 | 0.174 | 0.119 |
| SST | **−0.350** | **0.049** | **−0.362** | **0.034** |
| **Astrocytes*** | −0.136 | 0.101 | −0.164 | 0.085 |
| **Endothelial cells*** | 0.116 | 0.506 | −0.026 | 0.695 |
| **Oligodendrocytes*** | **0.199** | **0.049** | **−0.193** | **0.042** |
| **Microarchitectural axes** | ρ | q$_{spin}$ | ρ | q$_{spin}$ |
| Sensory-fugal cytoarchitectural differentiation | **0.294** | **0.009** | **0.369** | **0.004** |
| Anterior-posterior cytoarchitectural differentiation | 0.112 | 0.123 | 0.094 | 0.183 |
| Layer 1 thickness | **−0.219** | **0.009** | **−0.221** | **0.004** |
| Layer 2 thickness | −0.083 | 0.181 | −0.088 | 0.152 |
| Layer 3 thickness | 0.006 | 0.773 | −0.006 | 0.699 |
| Layer 4 thickness | **0.240** | **0.027** | **0.316** | **0.004** |
| Layer 5 thickness | −0.067 | 0.773 | −0.059 | 0.695 |
| Layer 6 thickness | **−0.157** | 0.052 | **−0.175** | **0.034** |

*Combined normalised gene expression for multiple cell-specific markers was used for overall cell categories. ρ: Spearman rank correlation. q$_{spin}$: *p*-value of correlation compared to 1000 spatially correlated spin permutations after correction of multiple comparisons for false discovery rate (corrected across all 24 comparisons for each group). In bold, statistically significant correlations.

differences emerged. In LBD, cortical organisational differences were concentrated in regions with reduced expression of genes related to signal transduction and response to stimuli. In PD-NC, differences were seen in regions with reduced expression of genes related to positive regulation of biological and cellular processes and catabolism.

Together, our findings demonstrate (1) the association between disease-related gradient differences and cortical cytoarchitecture,

neuronal populations and gene expression, (2) differences between PD-NC and LBD, suggesting distinct processes may underly cognitive and motor symptoms in Lewy body diseases and (3) the importance of multi-scale approaches integrating macroscale organisational principles with microarchitectural features for both validation and new hypothesis generation.

We found differences in cortical organisation in both PD-NC and LBD participants within the same regions, albeit regional changes were far more extensive in LBD. These alterations had different overall effects on global cortical hierarchies: In PD-NC inter-regional differences appeared to be amplified (consistent with changes illustrated in Fig. 1D), whereas in LBD overall gradient distribution was similar to controls, suggesting attenuation of regional specificity (Fig. 1F). This opposite effect on global cortical hierarchy, together with important differences in underlying cellular and molecular mechanisms, suggests that PD-NC and LBD diverge, at least partly, in underlying pathology rather than representing stages of the same pathological process, with LBD merely a more advanced state. The presence of terms uniquely enriched in relation to PD-NC, but not LBD organisational changes, further supports this. Our findings replicate, in living patients, observations about regional cortical specialisation from post-mortem PD and PDD brains[30]. Crucially, we show this divergent regional organisation between PD-NC and LBD at whole-brain level, using two different cohorts, imaging modalities and analysis techniques.

We replicate previous findings using single-cell transcriptomics, showing selective vulnerability of excitatory neurons in Lewy body dementias[30,33], particularly RORB-expressing excitatory neurons[30]. In LBD, cortical gradient differences were positively correlated with normative regional expression of RORB, a gene specifically expressed in layer 4 excitatory neurons[34], unlike in PD-NC. Because LBD mostly showed reduced gradient scores compared to controls, this implies that regions normally less rich in RORB (Fig. 4A) and with a thinner layer 4 (Fig. 4C) are more likely to change within the cortical hierarchy in LBD. Layer 4, the internal granular layer, varies considerably in thickness and composition across the cortex, and its small granular neurons are major postsynaptic targets of thalamic sensory nuclei, crucial for sensory processing[34,35]. Altered sensory processing, especially visual, is a cornerstone of Lewy body diseases, often happening early in the disease course or even predating symptoms[36,37] and correlating with cognitive impairment, predicting dementia in PD patients[5,38]. We found that reduced cortical differentiation compared to controls (lower overall SC-G1 gradient change score) was correlated with worse cognitive but not motor scores and only in LBD participants, suggesting these differences are specific to cognitive rather than motor impairment. Selective vulnerability of layer 4 neurons to pathological processes happening only in those patients with Lewy body disease with impaired cognition could potentially explain both the visuoperceptual deficits seen in Lewy body disease and why these deficits herald cognitive impairment[5,38].

There is some evidence that layer 4 vulnerability may be specific to alpha-synuclein pathology. Lewy bodies are more concentrated in

**Table 3 | Most significantly enriched GO biological terms**

| GO Biological Terms | | | | | | | |
|---|---|---|---|---|---|---|---|
| **Parkinson's with normal cognition (PD-NC)** | | | | **Lewy body dementias (LBD)** | | | |
| Term ID | Term name | Term size | *P*-value | Term ID | Term Name | Term size | *P*-value |
| GO:0007267 | Cell-cell signalling | 1316 | 8.17E-12 | GO:0044281 | Small molecule metabolic process | 1795 | 1.34E-17 |
| GO:0044281 | Small molecule metabolic process | 1795 | 9.39E-12 | GO:0019752 | Carboxylic acid metabolic process | 905 | 6.02E-12 |
| GO:0006810 | Transport | 4341 | 1.30E-10 | GO:0006810 | Transport | 4341 | 1.26E-11 |
| GO:0006631 | Fatty acid metabolic process | 398 | 3.12E-10 | GO:0043436 | Oxoacid metabolic process | 927 | 1.41E-11 |
| GO:0065008 | Regulation of biological quality | 2847 | 1.19E-09 | GO:0006082 | Organic acid metabolic process | 933 | 2.26E-11 |
| GO:0051049 | Regulation of transport | 1589 | 3.82E-09 | GO:0006631 | Fatty acid metabolic process | 398 | 7.75E-11 |
| GO:0010646 | Regulation of cell communication | 3453 | 4.01E-09 | GO:0051049 | Regulation of transport | 1589 | 5.43E-10 |
| GO:0019752 | Carboxylic acid metabolic process | 905 | 7.27E-09 | GO:0065008 | Regulation of biological quality | 2847 | 6.65E-10 |
| GO:0032879 | Regulation of localisation | 2005 | 8.08E-09 | GO:0007267 | Cell-cell signalling | 1316 | 1.67E-09 |
| GO:0023051 | Regulation of signalling | 3446 | 1.01E-08 | GO:0032879 | Regulation of localisation | 2005 | 6.79E-09 |

The GO and REAC term ID is a unique gene ontology identifier; the term name gives a brief description; and the term size is the number of unique genes associated with a given term. The reported P-values are corrected for multiple comparisons using the g:SCS algorithm in g:Profiler.
In bold terms that are missing in the other respective patient group.

cortical layers III-VI[39], and although a study comparing post-mortem visual cortex between AD and DLB did not show major laminar-specific differences, it did reveal functional changes within layer 4 GABAergic neurons[40]. Tau-pathology seems to spare layer 4 in Alzheimer's[41], with RORB-expressing layer 4 neurons particularly resilient to amyloid and tau pathology[42]. In contrast, gene expression changes within RORB-expressing excitatory neurons in PDD are explained by Lewy body but not amyloid pathological load[30]. Our findings further support this: regional gradient alterations in LBD aligned with the regional distribution of PD- but not AD-specific gene lists, and divergence from normative gradient ratings did not correlate with plasma p-tau levels. This suggests that cortical reorganisation in LBD is linked specifically to cognition and to alpha-synuclein pathology.

Whilst RORB layer 4 excitatory neurons may play a differential role in dementia in LBD, we also found important similarities between the drivers of gradient differences in LBD and PD-NC. In both groups, differences in cortical organisation followed the sensori-fugal axis, with primary sensory areas more downweighted, and this correlated with inhibitory neuron and oligodendrocyte regional expression. Impaired intracortical inhibition is well described in relation to motor symptoms in PD[43-45]. And while PD was traditionally considered as a purely neuronal disease, there is increasing evidence that oligodendrocytes play a significant role in early stages: transcriptomic changes in oligodendrocytes are repeatedly described in cortical and deep regions post-mortem[46-49] and white matter structural changes appear before or in the absence of grey matter loss in animal and cell models[50-52] and in vivo neuroimaging[11,12,49,53]. Our findings provide further evidence for this, though they do not indicate a specificity of oligodendrocyte involvement for cognitive impairment.

Whilst RORB layer 4 excitatory neurons were only correlated with LBD but not PD-NC organisational changes, we saw inhibitory neurons being correlated in both groups, albeit with some differences in underlying cell populations. Impaired inhibitory transmission is described across the spectrum of LBD: reduced GAD expression is seen in Parkinson's compared to controls[54], and reduced cortical GABAergic transmission is associated with hallucinations in DLB post-mortem[40] and Parkinson's in vivo[55]. However, there is less evidence regarding selective vulnerability in specific inhibitory neuronal populations. Whilst transcriptional changes have been seen in inhibitory neurons and other neuronal populations in PD[56] and PDD[30], a recent single-cell

transcriptomic study of the cingulate found inhibitory neurons to be less affected by cortical Lewy body pathology than excitatory neurons[33]. Our findings indicate that further investigation into the role of inhibitory neurons in PD with and without cognitive impairment is warranted.

Despite the striking structural organisational differences we saw in PD-NC and LBD compared to controls, we found no changes in functional gradients. This may reflect reduced power from a smaller sample size due to rsfMRI scans not passing quality control, or could indicate either more localised or compensatory changes. Functional connectivity compensation may play a role in delaying symptom onset during presymptomatic stages, and its loss may partially explain motor and cognitive symptom severity in Parkinson's[57-59]. Although not examined here directly, this could be clarified in future work.

The concordance of our in vivo results with post-mortem transcriptional alterations seen in PD and PDD[30] suggests that, at least partly, differences in cortical hierarchies revealed using gradient analyses reflect changes in gene expression in Lewy body diseases. It also provides important support for the use of multimodal and multi-scale approaches combining neuroimaging with normative gene expression and cytoarchitectural atlases. Whilst the value of such approaches in bridging scales and providing mechanistic insights into the healthy human brain is well-established[13,15,17], ongoing debate on their applicability in disease persists[60,61], not only due to methodological limitations but also because gene expression and cell composition are altered in neurodegeneration[30]. However, our findings suggest that these methods can yield important insights and can be used in a bidirectional way: not only can these methods be used to validate experiments at the cellular or molecular level, as we have done, but they can also generate new hypotheses and calibrate these experiments. For example, future transcriptomic studies comparing LBD and PD could target regions where we contribution of gene expression variability on cortical reorganisation was highest (PLS1 weightings, Supplementary Fig. 8A); whilst layer 4 specific differential changes in LBD versus Alzheimer's disease could target sensory areas and particularly the sensorimotor cortex which was also an area significantly downweighted in LBD (and thus correlated with reduced RORB expression and layer 4 thickness).

Our study has limitations. Our main aim was to assess differences in cortical organisation across the spectrum of Lewy body disease,

**Table 4 | Most significantly enriched REACTOME pathways**

**REAC pathways**

| Parkinson's with normal cognition (PD-NC) | | | | Lewy body dementias (LBD) | | | |
|---|---|---|---|---|---|---|---|
| Term ID | Term name | Term size | P-value | Term ID | Term Name | Term size | P-value |
| REAC:R-HSA-8978868 | Fatty acid metabolism | 173 | 7.50E-07 | REAC:R-HSA-8978868 | Fatty acid metabolism | 173 | 3.87E-06 |
| REAC:R-HSA-5661231 | Metallothioneins bind metals | 11 | 3.51E-04 | REAC:R-HSA-77289 | Mitochondrial Fatty Acid Beta-Oxidation | 37 | 8.45E-04 |
| REAC:R-HSA-77289 | Mitochondrial Fatty Acid Beta-Oxidation | 37 | 4.95E-04 | REAC:R-HSA-389887 | Beta-oxidation of pristanoyl-CoA | 8 | 9.19E-04 |
| REAC:R-HSA-389887 | Beta-oxidation of pristanoyl-CoA | 8 | 6.56E-04 | REAC:R-HSA-5661231 | Metallothioneins bind metals | 11 | 1.56E-03 |
| REAC:R-HSA-9033241 | Peroxisomal protein import | 60 | 8.84E-03 | **REAC:R-HSA-193368** | **Synthesis of bile acids and bile salts via 7alpha-hydroxycholesterol** | **23** | **4.44E-03** |
| REAC:R-HSA-5660526 | Response to metal ions | 14 | 9.17E-03 | REAC:R-HSA-8980692 | RHOA GTPase cycle | 149 | 1.35E-02 |
| REAC:R-HSA-390696 | Adrenoceptors | 9 | 1.26E-02 | **REAC:R-HSA-1430728** | **Metabolism** | **2079** | **1.36E-02** |
| **REAC:R-HSA-556833** | **Metabolism of lipids** | **735** | **2.14E-02** | REAC:R-HSA-9033241 | Peroxisomal protein import | 60 | 1.37E-02 |
| REAC:R-HSA-390918 | Peroxisomal lipid metabolism | 24 | 2.32E-02 | REAC:R-HSA-390696 | Adrenoceptors | 9 | 1.80E-02 |
| REAC:R-HSA-8980692 | RHOA GTPase cycle | 149 | 4.61E-02 | **REAC:R-HSA-164938** | **Nef-mediates down modulation of cell surface receptors by recruiting them to clathrin adaptors** | **21** | **2.16E-02** |

The GO and REAC term ID is a unique gene ontology identifier; the term name gives a brief description; the term size is the number of unique genes associated with a given term. The reported P-values are corrected for multiple comparisons using the g:SCS algorithm in g:Profiler.

In bold, terms that are missing in the other respective patient group.

particularly between those with and without dementia. However, these differences could be influenced by underlying alterations in subcortical-cortical connectivity, which we did not assess. Future work specifically examining subcortical-cortical organisation could address this. Secondly, we had limited sensitivity to functional gradient alterations due to participant-level censoring leading to loss of participant, short time duration and modest volumes; this could explain the absence of significant changes despite widespread structural changes. Thirdly, this was a cross-sectional study, and we could only compare organisational changes between groups. Future longitudinal work could characterise the timeline of changes across the spectrum of Lewy body diseases.

In summary, our study suggests that changes at the molecular level within the same regions are reflected in divergent alterations in cortical organisation in patients with Lewy body diseases with and without dementia. Despite similarities in cellular and molecular drivers of organisational differences, we also show important differences in patients with cognitive impairment, particularly in RORB layer 4 excitatory neurons; these are not driven by amyloid co-pathology and are specific to cognition. Finally, our study provides an important example of how multi-scale in vivo neuroimaging can be used to provide fundamental insights into the molecular underpinnings of neurodegeneration.

# Methods

## Participants

For both 3 T and 7 T cohorts, participants were recruited to University College London (UCL). All participants provided written informed consent, and the study was approved by the Queen Square Research Ethics Committee (15.LO.0476). The 3 T cohort has been previously described[5,11]. In brief, participants were required to be over 50 years of age, within 10 years of diagnosis, capable of providing informed consent and able to comply with study procedures. Additional inclusion criteria included:

- *Participants with Parkinson's disease (PD):* a clinical diagnosis in accordance with the Movement Disorders Society (MDS) clinical diagnostic criteria[62].
- *Participants with Dementia with Lewy Bodies (DLB)*: a clinical diagnosis of probable DLB in accordance with the McKeith diagnostic criteria[2].
- *Participants with Parkinson's disease dementia (PDD):* an established clinical diagnosis of Parkinson's disease and dementia or impaired function in activities of daily living (impaired functional assessments questionnaire) and Montreal Cognitive Assessment (MoCA) score below 26[3].
- *Participants with Parkinson's disease and mild cognitive impairment (PD-MCI):* a clinical diagnosis of PD[62] and persistent performance below 1.5 standard deviations (SD) in at least two different tests in one cognitive domain or one cognitive test in at least two cognitive domains, according to respective MDS criteria[63].
- *Control participants:* aged between 50 and 81.

Exclusion criteria included confounding neurological and psychiatric disorders and contraindications to MRI. Controls diagnosed with dementia, or mild cognitive impairment or Mini-mental State Examination (MMSE) score less than 25 were also excluded.

Participants with a diagnosis of DLB, PDD or PD-MCI were grouped as Lewy Body Dementia (LBD), with the remaining PD participants classified as PD-normal cognition (PD-NC).

**7 T cohort.** Due to stricter rules for 7 T MRI safety, we did not restrict participants with PD or LBD in terms of disease duration. Otherwise, diagnostic categories and inclusion-exclusion criteria were identical to those of the main 3 T cohort.

A total of 46 (*n* = 21 controls, *n* = 13 PD-NC and 13 LBD) participants with scans passing quality assurance (see *MRI quality control* below) were included. Of those participants, *n* = 23 (50%) also had 3 T scanning and are also included in the 3 T cohort. For those participants who had both a 3 T and 7 T scan, scanning for 7 T was performed between 1 and 4 years after their 3 T scan.

**Clinical assessments.** All participants underwent the same clinical and neuropsychological assessments. All assessments were performed with participants receiving their usual medications to minimise discomfort and minimise attrition. The Mini-Mental State Examination[64] (MMSE) and Montreal Cognitive Assessment[65] (MoCA) were used as measures of global cognition, and 2 tests per cognitive domain were performed[63]:

- *Attention*: Stroop colour naming task[66] and Digit span backwards[67]
- *Executive function*: Category fluency[68] and Stroop interference task[66]
- *Language*: Letter fluency[68] and Graded naming task[69]
- *Memory*: Word recognition task[70] and Logical memory task (delayed score)[67]
- *Visuospatial function*: Hooper visual organisation test[71] and Benton Judgement of line orientation[72].

A composite cognitive score was calculated as the average z-scores of MoCA plus one task per cognitive domain: inverted Stroop (colour naming time), Category fluency, Letter fluency, Word recognition task and Hooper Visual Organization Test, as we have previously described[5,11].

Global disease burden was assessed using the MDS United Parkinson's Disease Rating Scale (UPDRS) total score[73]. Motor severity was assessed using UPDRS part 3 (UPDRS-III)[73], and the timed up and go test (TUG)[74]. Cognitive fluctuations were measured using the clinician assessment of fluctuations (CAF)[75], one-day fluctuations scale[75] and dementia cognitive fluctuation scale (DCFS)[76]. Anxiety and depression were measured using the Hospital Anxiety and Depression Scale (HADS)[77]. Impairments in activities of daily living were measured using the Functional Activities Questionnaire[78]. Autonomic symptoms were assessed using the Compass-31 questionnaire[79]. Sleep disturbances were measured using the Rapid Eye Movement Behaviour Disorder Sleep Questionnaire (RBDSQ)[80] and visual hallucinations using the University of Miami PD hallucinations questionnaire (UMPDHQ)[81]. Levodopa equivalent daily doses (LEDD) were calculated for all PD-NC and LBD participants[82].

**Statistical analysis.** Between-group comparisons for demographics and clinical characteristics were performed using ANOVA for normally distributed and Kruskal–Wallis tests for non-normally distributed variables, with post-hoc t-test and Mann-Whitney, respectively. Shapiro-Wilk was used to determine normality.

## MRI data acquisition and quality control

All participants were scanned in the same scanners (3 T Siemens Prisma scanner for the main cohort and 7 T Siemens Terra scanner for the replication cohort) with the same protocol. Neuroimaging included:

Main cohort (3 T): structural T1-weighted scan (3D MPRAGE: magnetisation prepared rapid acquisition gradient echo), diffusion-weighted imaging (DWI) and resting-state functional MRI (rsfMRI).

T1 image was acquired with the following parameters: $1 \times 1 \times 1$ mm isotropic voxels, TE = 3.34 ms, TR = 2530 ms, flip angle = 7°, acquisition time ~6 min. DWI was acquired with: b0 (both AP and PA directions), $b = 50$ s/mm²/17 directions, $b = 300$ s/mm²/8 directions, $b = 1000$ s/mm²/64 directions, $b = 2000$ s/mm²/64 directions, $2 \times 2 \times 2$ mm isotropic voxels, TR = 3260 ms, TE = 58 ms, 72 slices, acceleration factor = 2, acquisition time ~10 min rsfMRI was acquired with: gradient-

echo EPI, TR = 3.36 s, TE = 30 ms, flip angle = 90°, FOV = 192 × 192, voxel size = 3 × 3 × 2.5 mm, 105 volumes, acquisition time ~6 min. During rsfMRI, participants were instructed to lie quietly with eyes open and avoid falling asleep (confirmed by monitoring and post-scan debriefing).

7 T cohort: structural T1-weighted scan (MP2RAGE) and multi-parameter mapping (MPM) acquisitions.

All 7 T data were acquired on the same 7 T Siemens Terra Scanner with a multi-echo variable flip angle MPM protocol[83,84] with whole-brain coverage and 0.6 mm isotropic voxel size. Two multi-echo 3D fast low-angle shot (FLASH) scans with proton density and T1-weighting were acquired with flip angles 6° and 24°, respectively, and readouts of alternating polarity to give 6 equidistant echoes with TE = 2.2–14.1 ms for PD and 2.3–14.2 ms for T1 and a common TR of 19.5 ms, FOV = 256 × 218 × 173 mm³, and readout bandwidth was 469 Hz/pixel. Parallel imaging was used with an acceleration factor of 2 in each phase and partition directions and 48 integrated reference lines. An additional multi-echo scan with magnetisation transfer weighting was acquired using the same settings as the PD weighted acquisition, but with just four echoes (2.2–9.34 ms). This scan included a 180° pre-pulse with 4 ms duration, 2 kHz off resonance. Calibration data to correct for transmit and receive field inhomogeneities were also acquired[85,86].

Parameter maps were then calculated using the hMRI toolbox in SPM12, using default toolbox configuration settings with correction for imperfect spoiling enabled[84], and the pre-computed B1 option. The following parameter maps were derived: proton density, sensitive to tissue water content, longitudinal relaxation rate (R1) sensitive to myelin, iron and water content, effective transverse relaxation rate (R2*) sensitive to iron content and less so myelin, and magnetisation transfer saturation (MTsat) particularly sensitive to myelin content.

All imaging sequences were performed with PD and LBD participants receiving their normal medications.

**MRI quality control**. Raw image data were visually inspected by two raters blinded to clinical data to ensure appropriate brain coverage and identify artefacts, e.g., motion. Further visual inspection was performed throughout the processing, including the final maps computed. In addition to visual inspection, we adopted strict motion-control criteria for rsfMRI, given susceptibility to motion artefact, similar to our previous work[29]. Only scans passing quality control for each modality were included. This resulted in a total of 111 participants included in the 3 T functional connectivity analyses (n = 23 controls, n = 35 PD-NC, and n = 50 LBD). Full details on MRI quality control and preprocessing of 3 T data are seen in Supplementary Methods.

**Cortical gradient construction**
**Gradient construction.** Following standard preprocessing of T1-weighted, DWI and rsfMRI data, using established pipelines in our group (see Supplementary Methods)[29], structural and functional connectomes were constructed for each participant using the same parcellation (Shaefer 200 cortical regions)[87]. Parcellations in the range of 200 nodes increase reliability in gradient construction[88], particularly those derived from functional connectivity. The same parcellation was used to construct both structural and functional connectomes, weighted by streamline count for structural and correlation coefficient for functional connectomes.

Cortical gradients were then derived separately from structural and functional connectivity matrices as well as from fused gradients using both structural and functional connectivity, by horizontally concatenating structural and functional connectivity matrices[89]. Gradient analyses were performed using BrainSpace[88]. Gradients were derived using diffusion map embedding, which identifies spatial axes of variation in connectivity across different brain regions, whereby cortical vertices that are strongly interconnected are closer together and vertices with little or no inter-connectivity are farther apart[17,90].

Normalised angle was used as a metric of similarity for this as it retains angular distance between regions and is less sensitive to noise (values range between 0 and 1, with 1 denoting identical angles, and 0 opposing angles). The normalised angle between two nodes i and j (A(i,j)) is calculated as in Equation (1):

$$A(i,j) = 1 - \frac{\cos^{-1}\left(cossim\left(x_i, x_j\right)\right)}{\pi}$$

where *cossim* is the cosine similarity function and x is the respective connectivity metric (streamline count for structural and Pearson correlation for functional connectivity matrices).

First, we generated a group-level gradient component template for each gradient type (structural, functional, fused), limiting the number of gradients to 10 and using default sparsity, thus keeping the top 10% of weights. Each group-level template was calculated from the average structural, functional and fused connectivity matrices of control participants (n = 23). Then individual gradients were derived from each participant's structural, functional and fused connectivity matrix (10 gradients, default sparsity 0.9, a = 0.5 diffusion). Procrustes alignment was then applied to the gradient components of each individual to align them to the group template; this enables gradients to be compared across individuals or groups[91]. Results were replicated using different sparsity parameters (sparsity 0.8 and 0.5).

**Statistical analysis.** Gradient scores were compared between groups (PD-NC vs HC, LBC vs HC and LBD vs PD-NC) using surface-based linear models implemented in BrainStat[92], with age and sex as covariates and family-wise error (FWE) correction using random field theory and the default cluster-defining threshold 0.01.

**Neural contextualisation**
We contextualised the differences in structural gradients found in LBD and PD-NC compared to HC with respect to normative variations in (1) excitatory and inhibitory neuronal gene expression markers, (2) cortical cytoarchitecture, (3) disease-specific genetic and plasma markers, and (4) global gene expression.

**Cell-specific gene expression.** The unthresholded t-map of differences in structural gradients (LBD vs HC and PD vs HC), corrected for age and sex, was parcellated (Shaefer 200 cortical regions) using the neuromaps Parcellater tool[93].

Gene expression profiles were obtained using data from the Allen Human Brain Atlas (AHBA)[32]. We extracted gene expression data and mapped them to 200 cortical regions using abagen and an established preprocessing pipeline[94,95]. Each tissue sample was assigned to an anatomical structure of the 200 cortical regions, using the AHBA MRI data for each donor. Data were pooled between homologous cortical regions to ensure adequate coverage of the left (six donors with data) and right hemisphere (two donors with data). Distances between samples were evaluated on the cortical surface with a 2 mm distance threshold. Probe-to-gene annotations were updated in Re-Annotator[96] with a background threshold of 50% of samples. A representative probe for a gene was selected based on the highest intensity. Gene expression data were normalised across the cortex using scaled, outlier-robust sigmoid normalisation. 15,745 genes (of 20,737 initially included) survived preprocessing. The resulting gene table was used to extract cell-specific gene markers and to assess global gene expression patterns.

Cell type-specific markers included:

- *Excitatory neurons*: CUX2, THEMIS, RORB, FEZF2
- *Inhibitory neurons*: PAX6, RELN, VIP, SV2C, TLE4, PVALB, SST
- *Astrocytes*: AQP4, GFAP
- *Endothelial cells*: CLDN5, SEMA3G, EFNB2, MFSD2A, SLC16A1, C1QA, HBB, ACTA2, CNN1, VWF, TSHZ2

- *Microglia*: CD74, PTPRC
- *Oligodendrocytes*: PDGFRA, BCAS1, PLP1, MBP, RASGRF1, ANKRD18A

Pooled gene expression for each of the above cell types was calculated per parcel by adding normalised gene expression values for each individual gene. Regional cell-specific gene expression was compared to the parcellated gradient difference t-map using Spearman correlation and spatial permutations (1000 spin permutations, statistical significance threshold $p_{spin} < 0.05$).

**Cytoarchitecture.** Histology-derived gradient maps reflecting axes of differentiation on cytoarchitecture in the healthy human brain and average regional laminar thickness values were obtained from BigBrain[97,98]; these are publicly available through BigBrainWarp[99]. Two main axes of differentiation were assessed: sensory-fugal (Histological gradient 1), differentiating between sensory and motor cortices from paralimbic structures and anterior-posterior (Histological gradient 2). We compared the parcellated t-map of gradient differences to these axes of cytoarchitectural differentiation, and to regional laminar thickness using Spearman correlation and spin permutation tests (1000 permutations, $p_{spin} < 0.05$).

**Association with disease severity and specificity.** To examine whether differences in cortical organisation are related to disease severity and specific for alpha-synuclein rather than driven by amyloid co-pathology, we performed two analyses. First, we calculated for each participant a composite gradient difference score. This was derived from the absolute Z-scored value for SC-G1 for each region compared to the control SC-G1 value for that region; this was then summed across the 200 regions to a single composite gradient difference score $Gs$ as seen in Equation (2):

$$Gs = \sum \frac{\left| x_{patient} - \mu\left( x_{controls} \right) \right|}{\sigma\left( x_{controls} \right)}$$

where x is the gradient value for a given region, μ is the mean and σ the standard deviation. Hence, higher Gs scores for a participant reflect a more different overall cortical organisation than controls. We correlated that score with outcomes of disease severity within each disease group (LBD and PD-NC) and with plasma p-tau217 using Spearman correlation.

Secondly, we examined whether the spatial distribution of LBD-related SC-G1 differences (unthresholded t-map of SC-G1 differences between LBD vs HC) was correlated with the spatial pattern of regional expression for Mendelian risk genes for PD and AD. We curated gene lists for Mendelian disease-associated genes from Blauwendraat et al[100] for PD (including only those classified as high or very high confidence for PD causation) and from OpenTargets for Alzheimer's disease, filtering for associated genes and requiring a genetic association score of >0.6. Our disease-specific gene lists are in Supplementary Methods. We then extracted summed normalised gene expression values for each of the 200 cortical regions of the Schaeffer parcellation from the processed AHBA gene expression table above. We used spatially correlated spin permutations (1000 permutations, $p_{spin} < 0.05$) to compare SC-G1 changes with regional disease-specific gene expression.

**Regional patterns of gene expression.** We used partial least squares regression (PLS) with predictor matrix X (the 200*15745 matrix of 200 regional mRNA measurements from 15475 genes extracted from AHBA[32] as described above) and dependent variable Y, the 200*1 regional structural gradient changes (t-map of gradient changes for LBD vs HC and PD vs HC separately). The first PLS component (PLS1), which explained the highest variance in gradient change and gene expression, was used to weigh and rank gene predictor variables.

1000 spin permutations[101] of the t-map of gradient changes was used to test the null hypothesis that PLS1 explained no more variance in Y than chance, and the null hypothesis of zero weight for each gene (q < 0.05, 1000 permutations based on sphere-projection-rotations[101]). Genes with significantly different weights than expected by chance were included in subsequent enrichment analyses, as in previous work[102–105].

**Enrichment analysis.** Gene Ontology (GO), Kyoto Encyclopedia of Genes and Genomes (KEGG) and Reactome pathway databases were used as references for enrichment. Enrichment analysis was performed using g:Profiler[106] with a significance threshold p < 0.05 (g:SCS method for multiple comparisons) and discarding terms associated with >2500 genes as being too general. The reduce and visualised gene ontology tool REViGO, based on semantic similarity, was used to visualise significant GO terms[107]. We used both Spearman's rank correlation and the hypergeometric test to assess term similarity between LBD and PD-NC enriched terms.

## 7 T region of interest analysis

MPM maps were generated using the hMRI toolbox[108]. Each participant's MP2RAGE image was co-registered to the proton density image, and all subsequent processing was performed in each participant's native proton density space. The same Schaffer parcellation (used for gradient construction in the 3 T dataset) was used to parcellate the brain into 200 cortical regions. Four regions of interest were selected for analysis based on their SC-G1 rankings and whether they showed differences or not in their ranking between LBD vs HC. These comprised two regions from the extremes of the gradient ranking, which showed differences in ranking between LBD and controls ("RH_SalVentAttn_TempOccPar_3", and "RH_SomMot_18") and two from the middle of the gradient distribution that did not show significant differences between LBD and HC ("RH_Default_Temp_1", "LH_Default_Temp_1"). For each ROI, we extracted the mean MPM signal for each of the MPM maps (R1, R2*, proton density and MTsat) and constructed general linear models with these as the dependent variables. The models included main effects of ROI and Group (LBD, PD-NC, HC) as well as their interaction. Age and sex were additionally included as covariates of no interest. The effect of interest was the ROI*Group interaction, to assess if the MPM values in the ROIs depended on having LBD.

## Replication and robustness

Several replication analyses were performed to ensure robustness of results:

- *Orthogonal analyses in 7T cohort using quantitative MRI*: We used an ultra-high field 7 T MRI cohort to validate differences in inter-regional differentiation in LBD and PD-NC than controls. This used different imaging sequences, partly different patients (50% of the 7 T cohort included in the main cohort, albeit with at least 1 year between the two scans) and different analysis methods, adding to the robustness of our results.

- *Gradient construction*: To ensure the choice of gradient construction did not influence results, we replicated our analyses with different sparsity options (sparsity 0.9 is presented in the main manuscript, and sparsity 0.8 and 0.5 are presented in Supplementary Results).

- *Spatial autocorrelation:* For all analyses of neural contextualisation, we used spatial permutations, based on spatially correlated sphere rotations[101], to test statistical significance. This ensures that correlations in the regional pattern of difference in cortical organisation (structural gradient differences seen in LBD vs controls and PD vs controls) and gene expression control for false positive bias due to gene-gene co-expression and spatial autocorrelation[109].

- *Enrichment analyses:* We only included significantly differentially weighted genes compared to spin permutations in enrichment analyses.

## Reporting summary

Further information on research design is available in the Nature Portfolio Reporting Summary linked to this article.

## Data availability

Source data are provided with this paper. Group-level data for demographics, clinical characteristics, and gradient scores are provided on: https://github.com/AngelikaZa/RegionalDifferentiationLBD. Individual-level neuroimaging data are under restricted access due to ethical reasons. Researchers interested in accessing the data should contact the senior author (r.weil@ucl.ac.uk) with a formal data request and research purpose, which will be reviewed in accordance with institutional policies. Please provide: (a) institutional affiliation; (b) a brief research proposal describing the intended analyses; (c) proof of ethical approval to perform the proposed research; and (d) a data security plan. Accepted users will need to sign an institutional formal data sharing agreement. Receipt of a request will be acknowledged within 10 business days, and a decision on whether or not we can proceed with a formal data sharing agreement will typically be issued within 60 calendar days. Source data are provided with this paper.

## Code availability

All processing and analysis code is available through: https://github.com/AngelikaZa/RegionalDifferentiationLBD, https://doi.org/10.5281/zenodo.17495352.

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

## Acknowledgements

We acknowledge all participants and their families who volunteered to take part in this study. A.Z. is supported by an Alzheimer's Research UK Clinical Research Fellowship (2018B-001) and grants from Parkinson's UK and Rosetrees Trust/Race Against Dementia. RSW is supported by a Wellcome Career Development Award (225263/Z/22). MGP is supported by the MS Society (Award reference: 183). This work was also supported by grants from the Lewy Body Society, Rosetrees Trust, an Academy of Medical Sciences Starter Grant, the National Institute for Health Research University College London Hospitals Biomedical Research Centre and the UK Dementia Research Institute UKDRI-2206 through UK DRI Ltd, principally funded by the Medical Research Council.

## Author contributions

A.Z. and R.S.W. conceived the study. AZ., G.T., M.G.P., M.R., M.F.C. and R.S.W. designed the study. A.Z., G.T., N.H. and I.D. collected and pre-processed the data. A.Z. analysed the data. A.Z. drafted the first version of the manuscript with input from G.T., M.G.P. and R.S.W. All authors reviewed and approved the final version of the manuscript.

## Competing interests

R.S.W. has received honoraria from GE HealthCare, Bial, Omnix Pharma and Britannia and consultancy fees from Therakind and Accenture, and UCL Partners. The other authors do not report any conflicts of interest.
