## [Transparent Peer Review file · Nature Communications]

Evidence for divergent cortical organisation in Parkinson's disease and Lewy Body Dementia

Corresponding Author: Dr Angeliki Zarkali

Version 0:

Reviewer comments:

Reviewer #1

(Remarks to the Author)

This study addresses the question of whether dementia associated with the Lewy body disease spectrum, encompassing Parkinson's disease and Lewy body dementia, is associated with altered gradients of brain structure and function. Structural and functional connectivity gradients were compared between healthy controls (HC; n=23) and two patient groups: Parkinson's disease with normal cognition (PD-NC; n=46) and Lewy body dementia (LBD; n=62). Relative to HCs, the two patient groups showed partially distinct patterns of alterations in a primary structural connectivity gradient, which the authors followed up with comprehensive analyses that assessed clinical correlations, overlap with specific cell populations, underlying microarchitecture, possible contributions of AD-like pathology, and regional gene expression. While the results are intriguing and the follow-up analyses are innovative, there are several issues that need to be addressed.

Major points

1. A main claim the authors make throughout the paper is that LBD and PD-NC show distinct alterations in structural connectivity gradients, which are subserved by distinct and shared alterations at the cellular and genetic levels. This claim, which is also in the title of the manuscript, is primarily based on findings from separate comparisons with healthy controls (i.e., LBD vs HC and PD-NC vs HC). However, as the authors report, there were no significant differences between LBD and PD-NC when these groups were compared directly, which would argue against the claim that these patient populations diverge in terms of underlying mechanisms of pathology. This needs to be discussed further, as it raises alternative interpretations of the results. For example, it may be argued that PD-NC and LBD share the same pathology, but that this pathology has not progressed to the same extent in PD-NC as it has in LBD. Such an interpretation would be consistent with the gradients differences displayed in figure 2.
2. In relation to the point above, it may be useful to carry out additional control analyses to test whether the LBD group, which consists of combination of PD with dementia (PDD; n=19) and dementia with Lewy bodies (DLB; n=36), is homogeneous or not with respect to the structural connectivity gradient.
3. Figures 2 and 3 include scatterplots that show the distribution of SC-G1 differences of LBD patients relative to controls. In many of these plots, points seem to gather into roughly bimodal distributions, characterized by two clusters appearing above or below a difference score of 0 relative to controls. Can the authors comment on what might be the cause of these distributions? This is a particularly pressing question for the correlation analysis presented in figure 2D, where a primary result of the paper looks to be driven by a small subset of patients who have highly negative difference scores. In relation to the point above, could these bimodal distributions be a consequence of combining PDD and DLB into a single group?
4. It would be useful to provide a more thorough explanation of what a gradient score entails, and how readers should interpret positive versus negative values or differences in scores between patients and controls. At times, the authors interpret patient-control differences in gradient scores using terms such as expansion, differentiation, contraction etc. without deeper explanation. However, it is not clear to the uninitiated reader what to make of these terms, which refer to re-organization of the entire gradient pattern, particularly in the context of the follow-up analyses. For example, in figure 2D, a negative gradient difference score is associated with better cognitive performance. Precisely what does such a negative difference score entail, and how can it support beneficial effects on cognition?

5. The authors may consider a different term than “replication” when describing their qMRI findings. These findings are derived using an entirely different analysis strategy (region-of-interest comparison) applied to a different neuroimaging modality. At best, this analysis can be said to provide additional support to the original findings but calling it a replication is perhaps too strong.

6. The introduction may be edited to clarify two main points. First, the aim of the study and its specific hypothesis about the similarity or divergence between diagnosis could be improved. Second, a more precise justification for using gradients to study this specific patient population would be helpful.

7. There is no clear discussion of the limitations of the study.

Minor points

1. The rs-fMRI preprocessing section in the supplement could be clarified further. For example, there is a large section detailing the CompCor strategy of deriving physiological noise regressors. However, in the paragraph below, it is stated that only 6 motion parameters together with CSF and WM signal was used for denoising. Is the CSF and WM signal derived from CompCor or not? Were the CompCor regressors used or not? If not, then it would be better to omit them from the manuscript.

2. The rs-fMRI denoising strategy employed in this study is sub-optimal. The authors may wish to consult the benchmarking study by Ciric et al. 2017 (<https://doi.org/10.1016/j.neuroimage.2017.03.020>) for alternatives. Poor denoising, combined with relatively long TRs and low number of volumes may explain the lack of findings in functional connectivity gradients. This might be mentioned as a limitation.

3. In figures 2-4, the color red is used to denote decreases relative to controls whereas blue is used to denote increases. This goes against convention, which is typically the reverse. It would be advisable to switch the color-coding scheme.

4. In figure 3, the regression lines in the scatterplots at the top do not have error estimates while all other regression lines do.

5. Supplementary table 2: it would be useful to know what the disease duration is for the different subgroups.

(Remarks on code availability)

Presently, the provided code is not sufficient to replicate the analyses in the manuscript. The supplied Jupyter Notebook provides insight into how the group comparisons of gradients were performed, and functions are provided that detail how those gradients were derived. However, there is no data available to test whether the code works as intended. Some data on structural connectivity gradients are provided, but this data does not include groupings of patients and controls. Moreover, the follow-up analyses of clinical correlations, microstructure etc. are not detailed in the code. Overall, more comments in the code and better instructions on their usage via the accompanying README file would be appreciated. The GitHub repository link referenced in the manuscript does not work, but this can be remedied at a later stage of submission.

Reviewer #2

(Remarks to the Author)

Summary:

This is a very well written and methodologically rigorous paper which shows expansion of structural connectivity gradients in patients with Parkinson's disease with normal cognition relative to healthy controls and patients with Dementia with Lewy Bodies or Parkinson's disease dementia. They contextualized these results by comparing group level difference maps to normative gene expression data from AHBA via neuromaps. The analysis focuses on the cortex due to methodology for computing spatial gradients and availability of molecular data. They found that structural abnormalities correlate with RORB expression which is a marker for layer 4 excitatory neurons, although they more strongly correlate with inhibitory interneuron markers. The 7T analysis does not really fit in well with the story but it is impressive that the authors have included so much multimodal data. Overall this is a strong paper that addresses the very difficult problem of linking quantitative human neuroimaging with neurobiology.

Major comments:

i. Can the authors provide a citation or further intuition for why more disparate gradient values indicated increased differentiation? They should also explain what exactly is meant by “differentiation” and what is the significance with regards to structural network organization and underlying neurobiology. Also, there are regions for which the gradient values are either higher or lower than controls in both groups, so it would be helpful to reconcile that finding with the interpretation of overall increased differentiation.

ii. The authors mention that functional connectivity results were not significant – can they include these results in the supplement?

iii. Can the authors provide justification for why the gradient analysis adds to the interpretation relative to comparing individual fiber strength or graph theory / connectivity metrics? For example, I'm not sure how to interpret the result that PD patients have increased PC1 values in the primary somatomotor cortices only on the left. Is the idea that these patients have a more isolated motor and visual cortex only on the left, but then a more integrated TPJ/parietotemporal cortex only on the right? The results with LBD are equally confusing. In short I wonder if the gradient analysis is aliasing a more interpretable analysis of individual connections or nodes. At the end of the day maybe the interpretation of the metric is not as important

as the fact that there are disease specific regional differences that correlate with cellular markers, but it would be nice to understand it a bit more.

iv. Can the authors provide an analysis to show that the correlation in Figure 2D is not driven by a few outlier patients with very low Z scored SC-G1 scores? Is there something different about these patients i.e. much more significant brain atrophy for other reasons such as cerebrovascular disease or copathology? The authors could also regress out plasma p tau values to ensure that does not explain the relationship between gradient values and cognition.

v. I don't understand how the results of the 7T analysis of Supplementary Figure 4 replicates findings in the remainder of the paper using the 3T analysis. These analyses seem different and the interpretation of the 7T metrics studied is not well explained. The authors should clarify and reinterpret accordingly. I think the 7T analysis should either be more complete (i.e. studying more than just the regions on the extremes of the gradient but look at all regions) or be the focus of a separate manuscript and not included here.

vi. The exclusion of subcortical connections seems like a major limitation given the importance of cortico-subcortical connections in PD/LBD where the disease is driven by pathology involving brainstem and basal ganglia structures before cortex. Can the authors comment on the choice to focus only on cortex and corticocortical connections in this work? Ideally if there is a way to include these connections in an analysis that would be great.

vii. The plasma p-tau analysis is an excellent control.

viii. The gene enrichment figure is not as interesting as the table – the terms identified and highlighted are very vague and don't shed light on any particular biologic process, compared with the table that is more specific. I wonder if the authors could save some space by moving this figure to the supplement.

ix. The cell type and cell specific marker expression analysis, as well as the laminar analysis, does not seem to have been corrected for multiple comparisons. This should be explicitly mentioned as the authors apply MC correction for other aspects of the paper but not here.

x. The authors focus primarily on RORB and oligodendrocytes in the discussion but the interneuron markers are more strongly correlated. Can they provide any explanation for the findings with regards to inhibitory interneurons?

(Remarks on code availability)

This repository is not publicly available so I cannot assess.

Reviewer #3

(Remarks to the Author)

Zarkali et al. have put together an extensive multimodal study comparing structural and functional organisational gradients in HC, LBD, and PD while also relating differences to performance on clinical scales and regional cell populations, cytoarchitecture, and gene expression. While the organisational gradient approach is intriguing, I found the measure difficult to put into a topological context due to the dimensionality reduction. Anchoring the concept more firmly in connectivity disruptions reported previously in the literature may help this understanding.

Please see additional comments below.

As I was unfamiliar with the concept of organisational gradients, I would appreciate if the authors could add more details on which type of spatial transitions are touched upon in the following sentence (II:72-74):

This large scale network architecture is underpinned by fundamental organisational gradients, which represent axes of continuous spatial transitions between brain regions.

For the replication cohort, the manuscript mentions that PD and LBD patients were not restricted regarding disease duration. Was there any disease duration-related restriction in place for the T3 cohort?

Were the parameter maps used for post-hoc connectome annotations?

How were structural and functional connectivity combined to produce fused gradients?

If I understand correctly, 50% of the patients in the 7T cohort were derived from the 3T cohort and not an entirely separate cohort.

VERY minor comments:

Please check sentence in II: 67-68

I would appreciate it if the MRI parameter section would follow the same sequence (T1, DWI, fMRI) as the preceding subheader.

(Remarks on code availability)

Version 1:

Reviewer comments:

Reviewer #1

(Remarks to the Author)

The authors took great care to respond to my previous comments and most of them are answered satisfactorily. They adequately alleviate my concerns about the distinctness of PDD and DLB. They also provide clear and transparent code to demonstrate how all quantitative stats were generated. Additionally, Figure 1 is a great addition to the manuscript that helps

to interpret gradient changes. I would like to request clarification on some of the responses:

- Reviewer 1, comment 3: I support taking the absolute difference in SC-G1 change. Given the aims of the paper, this makes sense. However, this does not answer my remark that the relationship between SC-G1 change and cognitive performance looks to be driven by a subset of LBD patients (mainly PDD and PD-MCI) who show much larger changes than the majority (mainly DLB). What could account for the fact that a small subset of both LBD and PD-NC groups show large changes whereas the remaining patients do not?

- Reviewer 1, comment 6: The additions to the introduction has made it very dense and somewhat difficult to read. It would be good to at least split the text into a new paragraph after "Our goal". The discussion is now also denser. The authors may consider working on the clarity and brevity of the text to save space and improve readability, as the manuscript now exceeds the word limit by a fair amount

- Reviewer 1, minor comments 1 and 2: I am confused as to whether global signal was removed or not. The italic text from comment 1 says that denoising involved "mean CSF, mean white matter, and global signals..." and that "we did not regress global signal". Do "global signals" refer to something different than "global signal"?

(Remarks on code availability)

The authors provide a fully transparent code base and a Jupyter Notebook that clearly demonstrates how all quantitative statistics were generated. They also provide gradient data. No clinical data is provided, which is understandable as these need to be protected. As far as I can tell there are no data of groupings that would allow users to verify differences in connectivity gradients between patient populations. Presently, only average connectivity gradients can be derived from the provided data. However, as noted, the Jupyter Notebook shows that if a user were provided with the clinical data, then said user would be able to fully reproduce the manuscript's statistics.

Reviewer #2

(Remarks to the Author)

Thanks to the authors for addressing my comments. I don't really have any additional substantive comments, other than to say I still think the terms differentiation and general discussion of gradients is flawed without an intuitive network, dynamical systems, or biologically based explanation of what gradients actually mean. Otherwise it is just a dimensionality reduction technique. I can interpret it then as an eigenvector of an LTI system, or generally in terms of how unimodal/transmodal a region is, but I am having trouble understanding how the shape of the distribution of gradient values is biologically interpretable and useful. It produces a biomarker that differs between these disease groups which can already readily be distinguished clinically, and it efficiently distills a matrix of connections into a vector.

(Remarks on code availability)

Yes - the code is available as a Jupyter notebook. It's a bit difficult to process as there's one notebook with hundreds of lines and multiple analyses. But if someone wanted to assess the code in great detail for errors, then they could.

Reviewer #3

(Remarks to the Author)

I thank the authors for having addressed all of my methodological comments.

(Remarks on code availability)

NCOMMS-25-53193-T: Evidence for divergent cortical organisation in Parkinson's disease and Lewy Body Dementia

We thank the editors for considering our manuscript for publication in *Nature Communications*. We were pleased that all reviewers found our paper **intriguing, innovative** and **well-described**.

We have now addressed all of the Reviewers' and Editor's comments. In particular, we performed separate analyses of different subgroups of Lewy body dementia (LBD) patients compared to controls (HC) and Parkinson's with normal cognition (PD-NC). We have clarified points in the methods, including a conceptual figure (Figure 1) to better illustrate gradient changes. We expand our discussion to include clearly described limitations and inhibitory neurons and have amended language throughout the manuscript to avoid the term replication for our 7T additional analyses. We provide a detailed response to each comment below.

We hope that the Editors and Reviewers agree that these changes have improved the manuscript and that it is now suitable for publication in *Nature Communications*.

Response to Reviewer 1:

Thank you for your thoughtful comments. We were pleased that the Reviewer found our manuscript intriguing and methods innovative. We have now addressed your comments below.

Major points

Comment 1: *"A main claim the authors make throughout the paper is that LBD and PD-NC show distinct alterations in structural connectivity gradients, which are subserved by distinct and shared alterations at the cellular and genetic levels. This claim, which is also in the title of the manuscript, is primarily based on findings from separate comparisons with healthy controls (i.e., LBD vs HC and PD-NC vs HC). However, as the authors report, there were no significant differences between LBD and PD-NC when these groups were compared directly, which would argue against the claim that these patient populations diverge in terms of underlying mechanisms of pathology. This needs to be discussed further, as it raises alternative interpretations of the results. For example, it may be argued that PD-NC and LBD share the same pathology, but that this pathology has not progressed to the same extent in PD-NC as it has in LBD. Such an interpretation would be consistent with the gradients differences displayed in figure 2."*

Response: We agree with the Reviewer that an alternative interpretation of the vertex-wise results could be that PD-NC and LBD share the same pathology but that this has not progressed to the same extent in PD-NC as in LBD.

However, there is some evidence from our data that this may not be the case. Firstly PD-NC and LBD participants show differences in global gradient distribution, with PD-NC alone showing gradient expansion, whilst overall gradient distribution in LBD participants was similar to controls. Secondly, whilst there are commonalities in the underlying cellular and

molecular pathways underpinning these changes, there are important differences between LBD and PD-NC: a) RORB4 (and layer 4 thickness) only correlate with organisational changes seen in LBD and b) there were pathways and gene ontology terms that were only enriched in one group and not the other. Specifically, the presence of 33 terms uniquely enriched in PD-NC but not LBD related changes is against LBD being a “more advanced” stage where identical pathology has just progressed to a greater degree.

We expand our Discussion to reflect this (Page 9):

“... This opposite effect on global cortical hierarchy, together with important differences in underlying cellular and molecular drivers of those organisational changes, suggest that these populations diverge, at least partly, in terms of underlying mechanisms of pathology. Additionally, the presence of terms uniquely enriched in relation to PD-NC but not LBD organisational changes goes against LBD reflecting merely a more advanced state of the disease, were the same pathological processes as in PD-NC are amplified. ...”

Additionally, we have now performed further subgroup analyses to assess different LBD subgroups compared to both controls and PD-NC. These are detailed under the response to Reviewer’s comment 2, below.

Comment 2. *“In relation to the point above, it may be useful to carry out additional control analyses to test whether the LBD group, which consists of combination of PD with dementia (PDD; n=19) and dementia with Lewy bodies (DLB; n=36), is homogeneous or not with respect to the structural connectivity gradient.”*

Response: We thank the Reviewer for raising this important point. To clarify whether the LBD group is homogeneous or not in respect to their structural connectivity gradient we separately compared DLB and PDD SC-G1 scores to controls (HC) and PD with normal cognition (PD-NC).

We found that both PDD and DLB participants showed qualitatively similar albeit less extensive changes in SC-G1 scores at the vertex level compared to controls (see *Supplementary Figure 2 below*). This suggests that the regional differences seen in LBD are not driven by any individual subgroup.

In addition, whilst overall SC-G1 distribution differed between subgroups within the LBD group (Kruskal Wallis $W=6.49$, $p=0.011$) there was no difference between PDD and DLB participants at post-hoc testing ($p=0.159$) with the only differences seen between PD-MCI group and DLB participants ($p<0.001$) and no difference between PD-MCI and PDD ($p=0.130$).

We report these findings in the main manuscript Results (Page 4):

“SC-G1 differences in LBD compared to controls were similar when assessing different LBD subgroups separately (Supplementary Figure 2).”

We also present these in full in *Supplementary Figure 2*.

Supplementary Figure 2: Individual subgroup SC-G1 changes compared to controls

Regional differences in SC-G1 scores between subgroups. Only statistically significant clusters after multiple comparisons correction ($p_{FWE} < 0.05$) are shown. Colour scale depicts effect size per vertex (blue colours decreases in gradient scores; red colours increases in gradient scores).

We additionally performed subgroup analyses between LBD subgroups (Dementia with Lewy bodies (DLB) and Parkinson's Dementia (PDD)) and PD-NC at vertex level, using surface-based linear models controlling for age and sex, family-wise error correction (FWE) for multiple comparisons and cluster threshold 0.01 assessed differences (as in our main comparisons).

There were no differences between PDD and PD-NC, but there were significant differences between DLB and PD-NC (see *Supplementary Figure 3* below). Importantly these were in the opposite direction than the differences seen between LBD and controls, PD-NC and controls (*Supplementary Figure 3*), or DLB and controls (*Supplementary Figure 2*). This suggests that this difference between DLB and PD-NC participants was not driving the results seen in our main analyses.

Comment 3: *“Figures 2 and 3 include scatterplots that show the distribution of SC-G1 differences of LBD patients relative to controls. In many of these plots, points seem to gather into roughly bimodal distributions, characterized by two clusters appearing above or below a difference score of 0 relative to controls. Can the authors comment on what might be the cause of these distributions? This is a particularly pressing question for the correlation analysis presented in figure 2D, where a primary result of the paper looks to be driven by a small subset of patients who have highly negative difference scores. In relation to the point above, could these bimodal distributions be a consequence of combining PDD and DLB into a single group?”*

Response: We apologise that this was not clear. The gradient difference score is meant to be an expression of how different overall cortical organisation is from normative cortical organisation; it is calculated as the sum of z-scores of gradient values for each cortical region against controls.

Negative and positive values are not necessarily of interest here, and we agree with the Reviewer that presenting them as such potentially hinders interpretability. Therefore, we have now amended the calculation of the composite gradient score to be the sum of the absolute value as below:

Composite gradient difference score G_s :

$$Gs = \sum \frac{|x_{patient} - \mu(x_{controls})|}{\sigma(x_{controls})}$$

Using the absolute value means that higher Gs values always reflect more different overall gradients than controls, which makes this measure more intuitive to interpret.

We have repeated our correlation analyses with measures of clinical severity and show that having gradient scores overall more similar to controls (lower Gs scores) was related to worse cognitive performance in LBD but not PD patients (see *Revised Figure 3, panel 3D presented below*). This provides further support that although overall inter-regional differentiation appears similar to control participants in LBD (with similar overall gradient distribution, *Revised Figure 3A*), there are underlying changes in organisation at the regional level (*Revised Figure 3B and C*) which are indeed behaviourally relevant; as participants with more “apparently normal” gradients are those with worse cognition.

Revised Figure 3D. Relationship between gradient organisational differences and composite cognitive scores.

As previously, there was no correlation between Gs scores and motor measures or levodopa equivalent daily dose, suggesting that these changes are specific to cognitive decline in Lewy body disease.

We have amended Figure 2 and our Results section (*Page 5*) with the new composite gradient difference score.

This relationship was mainly driven by less affected participants (PD-MCI and some PDD) showing overall difference in gradient scores compared to controls; in contrast, more cognitively affected PDD and DLB participants had lower Gs scores. Additionally, to ensure that no particular LBD subgroup was driving these results, we assessed whether Gs scores significantly differed between the LBD subgroups (PD-MCI, PDD, DLB). We found that

although overall the groups differed (Kruskal Wallis $W=27.1$, $p<0.001$) there were no significant differences between any of the subgroups in post-hoc testing (all individual comparisons showed $p>0.39$). We report this in *Supplementary Figure 8*.

Supplementary Figure 8. Differences in composite gradient difference score between

LBD subgroups (Left) and relationship with composite cognitive scores (Right).

Comment 4: “It would be useful to provide a more thorough explanation of what a gradient score entails, and how readers should interpret positive versus negative values or differences in scores between patients and controls. At times, the authors interpret patient-control differences in gradient scores using terms such as expansion, differentiation, contraction etc. without deeper explanation. However, it is not clear to the uninitiated reader what to make of these terms, which refer to re-organization of the entire gradient pattern, particularly in the context of the follow-up analyses. For example, in figure 2D, a negative gradient difference score is associated with better cognitive performance. Precisely what does such a negative difference score entail, and how can it support beneficial effects on cognition?”

Response: We apologise this was unclear. The gradient score reflects just the overall organisation of cortical hierarchies compared to controls. It doesn’t necessarily have a beneficial or adverse effect on cognition. We found that in people with dementia loss of difference compared to controls was accompanying cognitive decline, suggesting the changes we saw in overall organisation are behaviourally relevant.

We clarify this in our Results (Page 6):

“Next, we examined whether the changes in structural cortical organisation in Lewy body disease were related to disease severity. To do this we calculated for each participant, a composite gradient difference score, derived from the absolute Z-scored SC-G1 gradient value for each region compared to the control gradient values for that region, then summed across the 200 regions; this way higher scores for a participant reflect more differences in

overall cortical organisation than controls. We correlated composite gradient difference scores with measures of disease severity (cognitive severity: Montreal Cognitive Assessment (MOCA) and mini-mental state assessment (MMSE), and a composite cognitive score combining detailed cognitive assessments across 5 cognitive domains; and motor severity: Movement Disorders Society Unified Parkinson's Disease Scale part III (MDS-UPDRS-III) and "timed up and go" (TUG) score) within each disease group (LBD and PD-NC). We found that having gradient scores more similar to controls (lower composite gradient difference scores) was related to worse cognitive performance within LBD patients (SC-G1: MOCA: Spearman $\rho=0.234$, $p=0.062$; MMSE: $\rho=0.309$, $p=0.015$; composite cognitive score: $\rho=0.483$, $p<0.001$) but not within PD-NC (all non-significant; Figure 2D). This further supports the idea that apparent similarity in inter-regional differentiation between control and LBD participants is in fact associated with behaviourally relevant changes in organisation at the regional level. Indeed, participants with more "apparently normal" gradients are those with worse cognition."

And in our Discussion (Page 10):

"We saw that loss of difference in cortical organisation compared to controls (lower overall SC-G1 gradient change score) was correlated with worse cognitive but not motor scores and only in LBD participants but not in PD with normal cognition, suggesting these differences are specific to cognitive rather than motor impairment."

Comment 5: "The authors may consider a different term than "replication" when describing their qMRI findings. These findings are derived using an entirely different analysis strategy (region-of-interest comparison) applied to a different neuroimaging modality. At best, this analysis can be said to provide additional support to the original findings but calling it a replication is perhaps too strong."

Response: We agree with the Reviewer that the analyses methods of the 7T cohort are quite different, and the term replication may not be most appropriate. The rationale for these analyses were to provide orthogonal and independent support for the original findings by using different imaging modalities, different analyses and including partly different patient populations. We have now removed the term "replication" when referring to the 7T analyses throughout the manuscript.

For example in Results (Page 5):

"To ensure the robustness of our results, we provide further evidence in support of our findings in a separate cohort of LBD, PD-NC and controls using different MR acquisitions and analyses."

Comment 6: "The introduction may be edited to clarify two main points. First, the aim of the study and its specific hypothesis about the similarity or divergence between diagnosis could be improved. Second, a more precise justification for using gradients to study this specific patient population would be helpful."

Response: We apologise that this was unclear. We have expanded our Introduction to clarify how gradients may be useful specifically in Lewy body disease (Pages 2-3):

"Cortical gradients capture continuous cortical hierarchical changes^{14,24} and are thus more sensitive to global changes occurring in the presence of disease. Figure 1 illustrates how cortical gradients reflect inter-regional changes across different dimensions and how this

cortical organisation can change in the context of disease. Differences in these regional cortical organisation gradients have been found in psychiatric and neurological disease, with functional alterations described in both depression²⁵ and schizophrenia^{26,27} and structural alterations in patients with epilepsy²⁸. At the same time, these axes of organisation may be relevant to cognition, with structural gradient expansion, or increased spread between extremes of the cortical gradients (similar to changes seen in Figure 1C), reflecting higher inter-regional differentiation between unimodal and more transmodal regions in adolescence supporting executive function²⁹. In LBD where widespread, rather than focal alterations are commonly seen⁶⁻¹², assessing whether cortical gradients differ across the spectrum of the disease may be particularly relevant. At the same time, gradients offer a framework to explore underlying cellular and molecular correlates of such organisational changes^{14,24}.

We have also amended our Introduction to further clarify the aims and hypotheses (Page 3):

“Our goal was to answer this question in-vivo, assessing how large-scale differences in cortical structural and functional organisation differs between patients with LBD, PD-NC and unaffected controls (HC) (Figure 1) and explore their association with disease severity. Cortical gradients provide an ideal framework to assess this as they offer a measure of inter-regional differentiation sensitive to global organisational changes^{14,24}. We hypothesised that PD-NC would show increased spread between extremes of the gradient distribution (or overall gradient expansion, as illustrated in Figure 1D) reflecting more different connectivity profiles between regions^{28,29}, or increased inter-regional differentiation; whilst LBD would show similar overall distribution compared to controls but changes at regional and local level (similar to Figure 1E). Finally, we hypothesised that there would be differences in the underlying mechanisms of these changes between PD-NC and LBD patients.”

Comment 7: *“There is no clear discussion of the limitations of the study.”*

Response: We apologise that due to space constraints we had not included a distinct limitations section. We now include this in our Discussion (Page 11):

“Our study has several limitations. Our main aim was to assess differences in cortical organisation across the spectrum of Lewy body disease, particularly between those with and without dementia. However, these differences could be influenced by underlying alterations in subcortical-cortical connectivity which we were not able to assess within this work. Future work specifically examining subcortical-cortical organisation could address this. Secondly, we had limited sensitivity to functional gradient alterations which could explain the absence of significant changes despite widespread structural changes. We applied participant-level censoring leading to loss of participants and our rsfMRI data were limited by scan duration and thus included only a modest number of volumes. Thirdly, this was a cross-sectional study, and we could only compare organisational changes between groups. Future longitudinal work could characterise the timeline of changes across the spectrum of Lewy body diseases.”

Minor points

Comment 1: *“The rs-fMRI preprocessing section in the supplement could be clarified further. For example, there is a large section detailing the CompCor strategy of deriving physiological noise regressors. However, in the paragraph below, it is stated that only 6 motion parameters together with CSF and WM signal was used for denoising. Is the CSF and WM signal derived from CompCor or not? Were the*

CompCor regressors used or not? If not, then it would be better to omit them from the manuscript.”

Response: We apologise that the Supplement was confusing on this point. Although fMRIPrep computed CompCor components, we did not use them in denoising. Instead, we regressed the Friston-24 head-motion parameters (6 rigid-body, their first derivatives, and the squared terms) plus the mean CSF, mean white-matter, and global signals, each with first derivative and squared terms (total = 36 regressors). CSF and WM refer to mean tissue signals, not CompCor components. We have removed the detailed CompCor description to avoid implying that it was used:

“Sources of spurious variance were removed through linear regression including the Friston-24 head-motion parameters (trans/rot x,y,z; first derivatives; and their squared terms) and the mean CSF, mean white-matter, and global signals, each with first derivative and squared terms (total = 36 regressors). This was followed by calculation of bivariate correlations and application of Fisher transform. Given the contentiousness of global signal regression¹⁸ and potential to distort group differences¹⁹, we did not regress global signal.

Functional connectivity between ROIs was quantified as the Pearson correlation coefficient between mean regional BOLD time series resulting to a 200x200 undirected weighted connectivity matrix.”

Comment 2: *“The rs-fMRI denoising strategy employed in this study is sub-optimal. The authors may wish to consult the benchmarking study by Ciric et al. 2017 (<https://doi.org/10.1016/j.neuroimage.2017.03.020>) for alternatives. Poor denoising, combined with relatively long TRs and low number of volumes may explain the lack of findings in functional connectivity gradients. This might be mentioned as a limitation.”*

Response: Thank you for the pointer to Ciric et al., 2017. We implemented participant-level censoring by excluding high-motion participants (mean FD above threshold) in line with Parkes et al. 2018, who showed that a major benefit of censoring often derives from excluding high-motion individuals; this benefit significantly outperformed that of the choice of denoising strategy (Parkes Neuroimage 2018). We did not apply GSR because of its potential to alter distance-dependence and distort group differences even as it can reduce motion–FC associations (Ciric 2017, Parkes 2018). Our denoising strategy was a trade-off between removing motion artifacts and preserving degrees of freedom/signal as emphasised in both Ciric and Parkes.

Nevertheless, we agree that our limited volumes may reduce sensitivity of functional connectivity and gradient estimates. Unfortunately, having a limiting number of volumes was necessary in a clinical frail cohort to minimise scanning time. We now acknowledge this as a potential limitation in our Discussion (Page 11):

“we had limited sensitivity to functional gradient alterations which could explain the absence of significant changes despite widespread structural changes. We applied participant-level censoring leading to loss of participants and our rsfMRI data were limited by scan duration and thus included only modest volumes.”

Comment 3: *“In figures 2-4, the color red is used to denote decreases relative to controls whereas blue is used to denote increases. This goes against convention, which is typically the reverse. It would be advisable to switch the color-coding scheme”.*

Response: We have now inverted the colour-coding scheme in all figures in the main manuscript and Supplementary Material as per the Reviewer’s suggestion.

Comment 4: *“In figure 3, the regression lines in the scatterplots at the top do not have error estimates while all other regression lines do.”*

Response: We apologise for this error, the confidence intervals whilst plotted were not visible due to the size of scatter and line plots. We have now replotted Figure 3 to have visible error estimates (*Revised Figure 4*).

Comment 5: *“Supplementary table 2: it would be useful to know what the disease duration is for the different subgroups.”*

Response: We now include the overall disease duration (Years from diagnosis) in Supplementary Table 2. As expected, patients with Parkinson’s and mild cognitive impairment (PD-MCI) and those with Parkinson’s dementia (PDD) had longer disease durations than those with Dementia with Lewy bodies (DLB): PD-MCI, mean (standard deviation) = 8.8 (3.1); PDD 6.2 (4.9) and DLB 2.09 (1.8); Kruskal Wallis $H = 21.8$, $p < 0.001$.

Remark on code availability: *“Presently, the provided code is not sufficient to replicate the analyses in the manuscript. The supplied Jupyter Notebook provides insight into how the group comparisons of gradients were performed, and functions are provided that detail how those gradients were derived. However, there is no data available to test whether the code works as intended. Some data on structural connectivity gradients are provided, but this data does not include groupings of patients and controls. Moreover, the follow-up analyses of clinical correlations, microstructure etc. are not detailed in the code. Overall, more comments in the code and better instructions on their usage via the accompanying README file would be appreciated. The GitHub repository link referenced in the manuscript does not work, but this can be remedied at a later stage of submission.”*

Response: We thank the Reviewer for raising this point. The Jupyter Notebook for analysis is meant to illustrate the methodology of statistical comparisons. We also include code for deriving gradients which could be used in different datasets. We are not able to make publicly available individual level connectomes that could be used for individual level gradient construction, due to ethical constraints. However, to improve potential replication we now include patient groupings as well as age, sex and main clinical outcomes together with the code.

Clinical correlations are provided in the Jupyter notebook in the Subheading **“Correlations with clinical markers of severity”**. We have expanded the README file to include more instructions and details on the provided code and data.

The GitHub repository was still private on submission which is why the code was provided as a submission file. We have now made this public and the link should be live.

Response to Reviewer 2:

We thank the Reviewer for their thoughtful comments. We were pleased the Reviewer felt this was a “**very well written**” and “**methodologically rigorous**” paper. We address specific comments below.

Major comments:

Comment i: *“Can the authors provide a citation or further intuition for why more disparate gradient values indicated increased differentiation? They should also explain what exactly is meant by “differentiation” and what is the significance with regards to structural network organization and underlying neurobiology. Also, there are regions for which the gradient values are either higher or lower than controls in both groups, so it would be helpful to reconcile that finding with the interpretation of overall increased differentiation.”*

Response: We apologise this was unclear. At the global level, a widening of the cortical gradient can reflect a greater spread between extremes (e.g. transmodal vs unimodal for our structural gradient 1 SC-G1), therefore an exaggeration of the normally continuous hierarchy. Assessing the regions which are driving this change can reveal which regions are more affected by others increasing heterogeneity. We now include a conceptual figure (*Revised Figure 1*) that provides additional explanation on the concept of cortical gradients and what different changes in gradients may reflect.

Revised Figure 1. Conceptualisation of cortical gradients and potential alterations in the presence of disease.

A. Cortical gradients reflect continuous changes on different dimensions across the brain. Adapted from From Huntenburg et al. 2018. **Top:** Data points are coloured along the first dimension which is given as the distance between unimodal (sensorimotor) and transmodal regions (higher on the cortical hierarchy) along the cortical surface. When displayed on the cortical surface it aligns with a unimodal-to-transmodal cortical gradient. For any two regions, the difference in their gradient value represents how different these regions are in this dimension.

Bottom: Data points are coloured according to the second dimension which differentiates on this case between different sensory modalities. The position of each brain region along this dimension reflects its relative distance across three morphological landmarks (calcarine sulcus, transverse sulcus, central sulcus).

B-E: Conceptualisation of different possible changes in gradient distribution in the presence of disease.

The distribution of gradient values of a hypothetical structural connectivity gradient in healthy humans is shown in **B**. The distance between two brain regions (red and blue dots) reflects how different these brain regions are in their structural connectivity patterns along that cortical gradient dimension. In the presence of neurodegeneration, changes in structural connectivity may lead to alterations in gradient values between regions in different directions. **Gradient contraction (C)** reflects a more narrow distribution with more regions having gradient values along the mean. In this case, the difference between brain regions (line between the red and blue dots) is reduced. In contrast, widening of the overall gradient distribution, or **Gradient expansion (D)** with widening of extremes of gradient values, reflects increased differences between brain regions (red and blue dots) in terms of their structural connectivity profiles. Finally, whilst the overall distribution may remain similar to controls, there may be **Gradient reorganisation (E)** in the presence of disease. In this case brain regions change in their gradient value but the overall distribution remains the same.

Indeed, the global differences that we see in gradient distribution in PD-NC represent an increased spread between extremes of the gradient distribution (therefore heightened negative and positive values); this reflects increased dissimilarity between regions in their connectivity profiles (Royer 2023, Li 2024) or increased inter-regional differentiation (similar to the changes presented in *Revised Figure 1, panel D*). This is primarily driven by changes in unimodal regions.

In contrast in LBD patients, the overall gradient distribution was similar to the controls but there were widespread differences in spatial organisation at the local and regional levels (similar to the changes presented in *Revised Figure 1, panel E*).

We now clarify the interpretation of gradient expansion throughout the manuscript providing additional citations:

Introduction (Page 3):

“At the same time, these axes of organisation may be relevant to cognition, with structural gradient expansion, or increased spread between extremes of the cortical gradients, reflecting higher inter-regional differentiation between unimodal and more transmodal regions in adolescence supporting executive function²⁹”, and

“We hypothesised that PD-NC would show increased spread between extremes of the gradient distribution (or overall gradient expansion) reflecting more different connectivity profiles between regions^{28,29}, or increased inter-regional differentiation; whilst LBD would show similar overall distribution compared to controls but changes at regional and local level.”

Results (Page 4):

“W: we found that PD-NC patients showed expansion (scores shifting further away from the midpoint or increased spread between extremes of the distribution) of structural connectivity gradients compared to controls and LBD (SC-G1: Kruskal-Wallis $H=7.87$, $p=0.019$; SC-G2: $H=17.0$, $p<0.001$). This suggests that there is heightened dissimilarity in structural connectivity profiles between regions in PD-NC patients, or increased inter-regional differentiation between different brain regions.”

Comment ii: “The authors mention that functional connectivity results were not significant – can they include these results in the supplement?”

Response: We now include these in the Supplementary Material as Supplementary Figure 5.

Supplementary Figure 5. Differences in functional gradient scores between LBD, PD-NC and control (HC) participants

Comment iii: “Can the authors provide justification for why the gradient analysis adds to the interpretation relative to comparing individual fiber strength or graph theory / connectivity metrics? For example, I’m not sure how to interpret the result that PD patients have increased PC1 values in the primary somatomotor cortices only on the left. Is the idea that these patients have a more isolated motor and visual cortex only on the left, but then a more integrated TPJ/parietotemporal cortex only on the right? The results with LBD are equally confusing. In short, I wonder if the gradient analysis is aliasing a more interpretable analysis of individual connections or nodes. At the end of the day maybe the interpretation of the metric is not as important as the fact that there are disease specific regional differences that correlate with cellular markers, but it would be nice to understand it a bit more.”

Response: We apologise this was unclear. The main aim of the paper was to assess whether there are differences in global hierarchical organisation, and its underlying drivers, between LBD, PD-NC and controls. Cortical gradients are the ideal framework to assess this as they reflect a measure of continuous inter-regional differentiation, align well with known cortical hierarchies, evolutionary expansion and cytoarchitectonics, and they are sensitive to global organisational changes (Bernardt 2022, Wang 2025), which may be particularly important in LBD where widespread rather than focal changes are commonplace.

We have further explained our rationale for using cortical gradients in our Introduction (Pages 2-3):

“.... This large-scale network architecture is underpinned by fundamental organisational gradients, which represent axes of continuous spatial transitions between brain regions¹³. These continuous gradual changes in macroscale cortical organisation can be captured using eigenvector-based decompositions to cortex-wide similarity measures, or by calculating “cortical organisational gradients”. Cortical gradients allow us to link features of brain organisation across different scales and modalities¹⁴, and have been shown in health to underly brain structural^{15,16} and functional connectivity¹⁷, accompany brain development^{18,19} and evolution^{20,21} and also align with the brain’s regional cytoarchitecture and gene expression^{22,23}.

Cortical gradients capture continuous cortical hierarchical changes^{14,24} and are thus more sensitive to global changes occurring in the presence of disease. Differences in these regional cortical organisation gradients have been found in psychiatric and neurological disease, with functional alterations described in both depression²⁵ and schizophrenia^{26,27} and structural alterations in patients with epilepsy²⁸. At the same time, these axes of organisation may be relevant to cognition, with structural gradient expansion, or increased spread between extremes of the cortical gradients, reflecting higher inter-regional differentiation between unimodal and more transmodal regions in adolescence supporting executive function²⁹. In LBD where widespread, rather than focal alterations are commonly seen⁶⁻¹², assessing whether cortical gradients differ across the spectrum of the disease may be particularly relevant. At the same time, gradients offer a framework to explore underlying cellular and molecular correlates of such organisational changes^{14,24}.”

And:

“Our goal was to answer this question in-vivo, assess how large-scale differences in cortical structural and functional organisation differs between patients with LBD, PD-NC and unaffected controls (HC) (Figure 1) and explore their association with disease severity. Cortical gradients provide an ideal framework to assess this as they offer a measure of inter-regional differentiation sensitive to global organisational changes^{14,24}.”

Comment iv: “Can the authors provide an analysis to show that the correlation in Figure 2D is not driven by a few outlier patients with very low Z scored SC-G1 scores? Is there something different about these patients i.e. much more significant brain atrophy for other reasons such as cerebrovascular disease or copathology? The authors could also regress out plasma p tau values to ensure that does not explain the relationship between gradient values and cognition.”

Response: We thank the Reviewer for raising this point. Following the Reviewer’s comments, we have amended our composite gradient score to include absolute values (Gs, see Revised Figure 3D above) to increase interpretability, as the positive or negative direction does not carry much meaning here. The absolute values now mean that higher Gs values always reflect more different overall gradients than controls, which makes this measure more intuitive to interpret.

Following the Reviewer’s suggestion we additionally performed a linear regression, with age, sex and ptau-217 values (reflecting brain tau and also beta-amyloid co-pathologies) as covariates to ensure that none of these factors were driving the correlation between gradient values and cognition. We found that gradient values (Gs score) remained significantly correlated with combined cognitive scores ($p=0.004$) and MOCA ($p=0.045$) but not MMSE ($p=0.090$).

We report this in Results (Page 6):

“Correlation with gradient scores remained significant when correcting for age, sex and ptau-217 values for combined cognitive scores ($p=0.004$), MOCA ($p=0.045$) but not MMSE ($p=0.090$).”

Comment v: *“I don’t understand how the results of the 7T analysis of Supplementary Figure 4 replicates findings in the remainder of the paper using the 3T analysis. These analyses seem different and the interpretation of the 7T metrics studied is not well explained. The authors should clarify and reinterpret accordingly. I think the 7T analysis should either be more complete (i.e. studying more than just the regions on the extremes of the gradient but look at all regions) or be the focus of a separate manuscript and not included here.”*

Response: We apologise that this was not clear. The rationale for the 7T analyses were to provide orthogonal and independent support for the original findings by using different imaging modalities, different analyses and including a partly different patient populations. Our aim was to assess interregional differences between LBD, PD-NC and controls to show that these differ between groups in a similar way to our main analysis. It is not possible to assess inter-regional changes across all brain regions hence we adopted a region-of-interest approach. We focused on the regions at the extremes of the SC-G1 gradient (which differed between our groups in our main analysis) and compared these with regions in the middle of the distribution (which did not differ between groups). As such, this section compared quantitative MRI (qMRI) values between disease groups for each of these brain regions. We were particularly interested in myelin-sensitive markers (MTsat and R1), given our gradient changes were seen in structural connectivity gradients.

We agree that the term replication is not the most appropriate and have now removed it throughout the manuscript, and instead refer to it as a supportive analysis. Additionally, we expand our rationale and hypotheses for this analysis in the Introduction (Pages 3-4):

“We provide further evidence to support our main findings using region-of-interest (ROI) analysis and 7T quantitative MRI (qMRI) in a separate cohort; specifically we assessed whether regions at the extremes of SC-G1 rankings differed more or less in terms of their connectivity profiles between LBD, PD-NC and controls. We confirmed that inter-regional differences in mean qMRI values for myelin sensitive metrics alone across regions of different SC-G1 rankings, significantly differed between groups.”

And Results (Page 5):

“We were particularly interested in myelin-sensitive markers (MTsat and R1), given our gradient changes were seen in structural connectivity gradients.”

Comment vi: *“The exclusion of subcortical connections seems like a major limitation given the importance of cortico-subcortical connections in PD/LBD where the disease is driven by pathology involving brainstem and basal ganglia structures before cortex. Can the authors comment on the choice to focus only on cortex and corticocortical connections in this work? Ideally if there is a way to include these connections in an analysis that would be great.”*

Response: We agree with the Reviewer that subcortical connectivity alterations play a key role in PD and LBD. However, the primary aim of this study is to assess changes in cortical (rather than subcortical) organisation; this is the reason we chose cortical gradients as an analysis method given they reflect the global hierarchical organisation of the cortex. Cortical reorganisation could indeed be influenced by changes in subcortical-cortical connectivity but we are unable to directly assess this within this work. We now highlight this in our Discussion (Page 10):

“The main aim of our study was to assess differences in cortical organisation across the spectrum of Lewy body disease, particularly between those with and without dementia. However these differences could be influenced by underlying alterations in subcortical-cortical connectivity which we were not able to assess within this work. Future work specifically examining subcortical-cortical organisation could address this.”

Comment vii: *“The plasma p-tau analysis is an excellent control.”*

Response: We are pleased that the Reviewer agreed the plasma p-tau analysis provided good evidence against these changes being driven by amyloid co-pathology.

Comment viii: *“The gene enrichment figure is not as interesting as the table – the terms identified and highlighted are very vague and don’t shed light on any particular biologic process, compared with the table that is more specific. I wonder if the authors could save some space by moving this figure to the supplement.”*

Response: We thank the Reviewer for raising this point. We have now moved Figure 4 to the Supplementary Material as Supplementary Figure 8.

Comment ix: *“The cell type and cell specific marker expression analysis, as well as the laminar analysis, does not seem to have been corrected for multiple comparisons. This should be explicitly mentioned as the authors apply MC correction for other aspects of the paper but not here.”*

Response: We thank the Reviewer for highlighting this. We now have corrected for multiple comparisons for gradient changes for each group (PD-NC and LBD) using false-discovery rate, after spin-permutation. We have amended Figure 3 and Table 2 to now report this FDR-corrected p_{spin} value (q_{spin}).

Comment x: *“The authors focus primarily on RORB and oligodendrocytes in the discussion but the interneuron markers are more strongly correlated. Can they provide any explanation for the findings with regards to inhibitory interneurons?”*

Response: We primarily focused our discussion on neurons on RORB4 for two reasons: firstly, this was different between PD-NC and LBD patients (with only changes in LBD patients showing a correlation with RORB4 expression) and secondly, this finding was also seen in single-cell RNA sequencing at post-mortem.

However taking into account the Reviewer’s comments, we now include a paragraph in our discussion on inhibitory interneurons:

“Whilst RORB layer 4 excitatory neurons were only correlated with LBD but not PD-NC organisational changes, we saw inhibitory neurons being correlated in both groups, albeit with some differences in underlying populations driving these changes. Evidence of impaired inhibitory transmission has been previously described across the spectrum of LBD: reduced GAD expression is seen in Parkinson’s compared to controls⁵⁵ and reduced cortical GABAergic transmission has been seen in relation to hallucinations in DLB post-mortem⁵⁶ and Parkinson’s in-vivo⁵⁷. However, there is less evidence regarding selective vulnerability in specific inhibitory neuronal populations. Whilst transcriptional changes have been seen in inhibitory neurons, as well as other neuronal populations in patients with PD⁵⁸ and PDD³², in another recent single-cell transcriptomic study of the cingulate, inhibitory neurons were found to be less frequently affected by cortical Lewy body pathology than excitatory neurons³⁵. Our findings indicate that further investigation into the role of inhibitory neurons in PD with and without cognitive impairment is warranted.”

Remark on code availability: *“This repository is not publicly available so I cannot assess.”*

Response: As the repository was private at the time of submission, we provided the relevant code in a separate zip file. We have now made the repository publicly available and have updated it to include the more recent analyses performed during the manuscript’s Revision.

Response to Reviewer 3.

We thank the Reviewer for their useful feedback. We respond to specific comments below:

Comment: *“While the organisational gradient approach is intriguing, I found the measure difficult to put into a topological context due to the dimensionality reduction. Anchoring the concept more firmly in connectivity disruptions reported previously in the literature may help this understanding.”*

Response: We apologise this was not clear. Our structural connectivity gradient SC-G1 reflects how different brain regions differ or are similar in regard to their structural connectivity profiles. As such, in the presence of neurodegeneration, changes in structural connectivity within brain regions will lead to changes in organisational gradients. In LBD, we have previously shown widespread changes in structural connectivity (Zarkali 2024; Hannaway 2025); our current findings of regional re-organisation in LBD are in keeping with previously reported connectivity disruptions.

We now further explain the concept of organisational gradients and how this may change in the presence of disease in our *Revised Figure 1*. Additionally we specify why assessing changes in global organisation, using cortical gradients, may be particularly of value in LBD in our Introduction (Pages 2-3):

“Several studies have shown widespread changes in brain structure in association with cognitive decline in Lewy body diseases, including cortical atrophy^{6,7} and wide spread changes in grey⁸⁻¹⁰ and -white matter^{11,12} macrostructure. Although these studies provide useful insights, they were unable to take into account how different brain regions are interlinked at the cellular and molecular level across large-scale brain networks.”... “In LBD where widespread, rather than focal alterations are commonly seen⁶⁻¹², assessing whether cortical gradients differ across the spectrum of the disease may be particularly relevant. At the same time, gradients offer a framework to explore underlying cellular and molecular correlates of such organisational changes^{14,24}.”

Comment: “As I was unfamiliar with the concept of organisational gradients, I would appreciate if the authors could add more details on which type of spatial transitions are touched upon in the following sentence (ll:72-74):

This large scale network architecture is underpinned by fundamental organisational gradients, which represent axes of continuous spatial transitions between brain regions.”

Response: We thank the Reviewer for raising this point. We now include an explanatory Figure (*Revised Figure 1*) that further explains the concept of cortical gradients and what possible alterations in the presence of neurodegeneration may signify.

We also expand our Introduction to include more details on cortical gradient construction (Pages 2-3) and their interpretation:

“... This large-scale network architecture is underpinned by fundamental organisational gradients, which represent axes of continuous spatial transitions between brain regions¹³. These continuous gradual changes in macroscale cortical organisation can be captured using eigenvector-based decompositions to cortex-wide similarity measures, or calculating “cortical organisational gradients”. Cortical gradients allow us to link features of brain organisation across different scales and modalities¹⁴, and have been shown in health to underly brain structural^{15,16} and functional connectivity¹⁷, accompany brain development^{18,19} and evolution^{20,21} and also align with the brain’s regional cytoarchitecture and gene expression^{22,23}.

Cortical gradients capture continuous cortical hierarchical changes^{14,24} and are thus more sensitive to global changes occurring in the presence of disease. Differences in these regional cortical organisation gradients have been found in psychiatric and neurological disease, with functional alterations described in both depression²⁵ and schizophrenia^{26,27} and structural alterations in patients with epilepsy²⁸. At the same time, these axes of organisation may be relevant to cognition, with structural gradient expansion, or increased spread between extremes of the cortical gradients, reflecting higher inter-regional differentiation between unimodal and more transmodal regions in adolescence supporting executive function²⁹. In LBD where widespread, rather than focal alterations are commonly seen⁶⁻¹², assessing whether cortical gradients differ across the spectrum of the disease may be particularly relevant. At the same time, gradients offer a framework to explore underlying cellular and molecular correlates of such organisational changes^{14,24}.”

Comment: “For the replication cohort, the manuscript mentions that PD and LBD patients were not restricted regarding disease duration. Was there any disease duration-related restriction in place for the T3 cohort?”

Response: We apologise this was not clear in the main manuscript. The 3T cohort was restricted to 10 years disease duration (from diagnosis). We now clarify this in Methods (Page 11):

“The 3T cohort has been previously described^{5,11}. In brief, participants were required to be over 50 years of age, within 10 years from diagnosis, capable of providing informed consent and able to comply with study procedures...”

Comment: “Were the parameter maps used for post-hoc connectome annotations?”

Response: No, the parameter maps were only used in region of interest analysis to support our main findings.

Comment: *“How were structural and functional connectivity combined to produce fused gradients?”*

Response: Structural and functional connectivity gradients were combined following the approach described by Paquola et al (Plos Biology 2020), where structural and functional connectivity matrices were similarly integrated. We used horizontal concatenation of structural and functional connectivity matrices and then produced a node-to-node affinity matrix using row-wise normalised angle similarity. The fused affinity matrix thus quantifies the strength of cortical wiring between 2 regions.

We have clarified this in our Methods (Page 15):

“Cortical gradients were then derived separately from structural and functional connectivity matrices as well as from fused gradients using both structural and functional connectivity, by horizontally concatenating structural and functional connectivity matrices as previously described⁸⁶.”

We also include the function used to calculate fused gradients in our code (GetSubjectGradients.py).

Comment: *“If I understand correctly, 50% of the patients in the 7T cohort were derived from the 3T cohort and not an entirely separate cohort.”*

Response: Indeed, these are not entirely separate cohorts. However, we feel that the 7T analyses provides strong independent evidence to support our main findings for three reasons: 1) at least 1 year of time passed between the 3T and 7T scans, 2) only 50% of the patients in the 7T cohort (23/46) and 17.6% of the patients in the 3T cohort (23/131) were included in both cohorts and 3) the imaging acquisitions and analyses methods were different.

We clarify this further in Methods: Replication and robustness (Page 17):

“...We used a cohort of ultra-high field 7T MRI to validate differences in inter-regional differentiation in LBD and PD-NC than controls. This used different imaging sequences, partly different patients (50% of 7T cohort was also included in the 3T cohort, albeit with at least 1 year between the two scans) and different analysis methods, further adding to the robustness of our results.”

Comment: *“Please check sentence in II: 67-68. I would appreciate it if the MRI parameter section would follow the same sequence (T1, DWI, fMRI) as the preceding subheader.”*

Response: We have amended this paragraph to follow the same order, as per the Reviewer’s suggestion (Page 14):

“Main cohort (3T): structural T1-weighted scan (3D MPRAGE: magnetization prepared rapid acquisition gradient echo), diffusion-weighted imaging (DWI) and resting state functional MRI (rsfMRI).

T1 image was acquired with the following parameters: 1×1×1 mm isotropic voxels, TE=3.34 ms, TR=2530 ms, flip angle=7°, acquisition time ~6min. DWI was acquired with: b0 (both AP and PA directions), b=50 s/mm²/17 directions, b=300 s/mm²/8 directions, b=1000 s/mm²/64 directions, b=2000 s/mm²/64 directions, 2×2×2 mm isotropic voxels, TR=3260 ms, TE=58 ms, 72 slices, acceleration factor =2, acquisition time ~10 min rsfMRI was acquired with: gradient-echo EPI, TR=3.36 s, TE=30 ms, flip angle=90°, FOV=192×192, voxel size = 3×3×2.5 mm, 105 volumes, acquisition time ~6 min. During rsfMRI, participants were instructed to lie quietly with eyes open and avoid falling asleep (confirmed by monitoring and post-scan debriefing).”

NCOMMS-25-53193A: Evidence for divergent cortical organisation in Parkinson's disease and Lewy Body Dementia

We thank the editors for provisionally accepting our manuscript for publication in *Nature Communications*. We were pleased that all reviewers found our responses to their comments satisfactory and methodological concerns addressed.

We have now clarified and address all of the Reviewers' remaining comments. We also have addressed the Editorial requests on the separate document provided.

We hope that the Editors and Reviewers agree that these changes have improved the manuscript and that it is now suitable for publication in *Nature Communications*.

Response to Reviewer 1:

We were pleased the Reviewer found most of our responses satisfactory and their concerns regarding PDD and DLB groups alleviated. We now further clarify their remaining comments below:

Reviewer 1, comment 3: *"I support taking the absolute difference in SC-G1 change. Given the aims of the paper, this makes sense. However, this does not answer my remark that the relationship between SC-G1 change and cognitive performance looks to be driven by a subset of LBD patients (mainly PDD and PD-MCI) who show much larger changes than the majority (mainly DLB). What could account for the fact that a small subset of both LBD and PD-NC groups show large changes whereas the remaining patients do not?"*

Response: We thank the Reviewer for this comment. The subset of patients that are showing the greater changes in SC-G1 composite scores compared to controls, are the least cognitively affected with changes similar to that of the PD-NC participants. This is in keeping with an increased inter-regional differentiation in PD-NC that is attenuated with increased cognitive severity in LBD.

To ensure that there was no factor confounding this correlation we tested whether the LBD participants that showed these larger change scores differ from remaining LBD participants in non-cognitive metrics.

The LBD participants with greater changes in SC-G1 composite scores did not differ from the other LBD participants in demographics or motor characteristics:

Characteristic	Statistic	p-value
Age	H=1.90	0.168
Sex	$\chi^2=0$	1.0
Years education	H= 0.80	0.371
Years diagnosis	H =1.98	0.159
UPDRS total	H=1.86	0.173
UPDRS motor	H=0.10	0.752
LEED	H=0.43	0.510

We now report this additional analysis in *Supplementary Figure 8*.

Reviewer 1, comment 6: *“The additions to the introduction has made it very dense and somewhat difficult to read. It would be good to at least split the text into a new paragraph after “Our goal”. The discussion is now also denser. The authors may consider working on the clarity and brevity of the text to save space and improve readability, as the manuscript now exceeds the word limit by a fair amount”*

Response: We agree with the Reviewer that the additions to the introduction and discussion, in response to Reviewers’ comments, have meant that the manuscript exceeds the word limit. We split the text in the Introduction as per the Reviewer’s comment and we have refined the Manuscript, particularly the Introduction and Discussion sections, as much as possible to reduce word count and improve readability whilst maintaining all the additions that Reviewers requested. We now have cut down further 368 words, bringing the total word count to 4939 words, and improving readability.

Reviewer 1, minor comments 1 and 2: *“I am confused as to whether global signal was removed or not. The italic text from comment 1 says that denoising involved “mean CSF, mean white matter, and global signals...” and that “we did not regress global signal”. Do “global signals” refer to something different than “global signal”?*

Response: We thank the Reviewer for flagging this inconsistency in the Supplementary Material. We used the Friston-24 head-motion parameters to regress confounds which did include global signal. We removed the conflicting sentence from the Supplementary:

“Sources of spurious variance were removed through linear regression including the Friston-24 head-motion parameters (trans/rot x,y,z; first derivatives; and their squared terms) and the mean CSF, mean white-matter, and global signals, each with first derivative and squared terms (total = 36 regressors). This was followed by calculation of bivariate correlations and application of Fisher transform.”

Reviewer #1 (Remarks on code availability): *“The authors provide a fully transparent code base and a Jupyter Notebook that clearly demonstrates how all quantitative statistics were generated. They also provide gradient data. No clinical data is provided, which is understandable as these need to be protected. As far as I can tell there are no data of groupings that would allow users to verify differences in connectivity gradients between patient populations. Presently, only average connectivity gradients can be derived from the provided data. However, as noted, the Jupyter Notebook shows that if a user were provided with the clinical data, then said user would be able to fully reproduce the manuscript's statistics.”*

Response: We apologise for this omission, caused by an error in committing the group level files that went undetected. We now supply anonymised clinical information (including groupings, demographics and results of the main clinical assessments used in the paper) in our github repository under DATA.

Response to Reviewer 2:

We were pleased that the Reviewer was satisfied with our responses and they had no additional substantive comments. We clarify their minor comment below.

Comment: *“Thanks to the authors for addressing my comments. I don't really have any additional substantive comments, other than to say I still think the terms differentiation and general discussion of gradients is flawed without an intuitive*

network, dynamical systems, or biologically based explanation of what gradients actually mean. Otherwise it is just a dimensionality reduction technique. I can interpret it then as an eigenvector of an LTI system, or generally in terms of how unimodal/transmodal a region is, but I am having trouble understanding how the shape of the distribution of gradient values is biologically interpretable and useful. It produces a biomarker that differs between these disease groups which can already readily be distinguished clinically, and it efficiently distills a matrix of connections into a vector.”

Response: Thank you for your comment. We appreciate how gradients can be seen as simply a dimensionality reduction tool. However, this is not the case, and gradients are ideally suited to answer our study aim.

Our primary goal was to assess in vivo how large-scale cortical structural and functional organisation differs in patients with LBD, PD-NC and healthy controls. We aimed to replicate, in-vivo and at whole-brain level results from post-mortem single cell transcriptomics showing increased inter-regional differentiation in PD and attenuation of inter-regional differentiation in PDD (Fairbrother-Browne 2025).

Cortical gradients provide an ideal framework for this because, by definition, they capture inter-regional differentiation and are sensitive to global organisational shifts. Therefore, when we see increased spread between extremes of the gradient distribution in PD-NC this suggests greater differentiation between regions in terms of connectivity profiles (regions diverging, as illustrated in Figure 1D). When we see a compressed or normalised distribution as in LBD, this suggests reduced differentiation or a loss of hierarchical segregation (regions becoming more similar, as illustrated in Figure 1E). This has biological plausibility: in neurodegeneration, specialisation may be lost and regions may degrade toward more uniform connectivity profiles.

Importantly, gradients are not just mathematical constructs: they map onto biological, developmental and functional processes. For example, the same cortical gradients are seen cross species (Valk 2020, Valk 2022) and during development (Xia 2022; Dong 2021), whilst in adolescence, structural gradient expansion (greater spread between unimodal and transmodal regions) has been shown to support executive function (Li 2024).

We further link the gradient metrics to cognitive severity, and to underlying cellular/molecular correlates (in our study), which supports the interpretability beyond pure dimensionality reduction.

We have now simplified our Introduction and Discussion to clarify this point, whilst reducing on the word count and readability.

Reviewer #2 (Remarks on code availability): *“Yes - the code is available as a Jupyter notebook. It's a bit difficult to process as there's one notebook with hundreds of lines and multiple analyses. But if someone wanted to assess the code in great detail for errors, then they could.”*

Response: We have further amended the Readme file to better explain the different code and data provided. We have also annotated the main notebook with subsections which would make it easier for a reader to navigate.

Response to Reviewer 3:

We were pleased that the Reviewer was satisfied with our responses.